

# Dual-hemisphere sea ice thickness reference measurements from multiple data sources for evaluation and product inter-comparison of satellite altimetry

Ida Birgitte Lundtorp Olsen[1,9,*], Henriette Skourup[1,*], Heidi Sallila[2], Stefan Hendricks[3], Renée Mie Fredensborg Hansen[1,4], Stefan Kern[5], Stephan Paul[3], Marion Bocquet[6], Sara Fleury[6], Dmitry Divine[7], and Eero Rinne[8]

[1]Department of Geodesy and Earth Observation, National Space Institute, Technical University of Denmark (DTU Space), Elektrovej Building 327, 2800 Kgs. Lyngby, Denmark
[2]Marine Research Unit, Finnish Meteorological Institute (FMI), Helsinki, Finland
[3]Alfred Wegener Institute (AWI), Helmholtz Centre for Polar and Marine Research, Bremerhaven, Germany
[4]Department of Civil and Environmental Engineering, Norwegian University of Science and Technology (NTNU), Trondheim, Norway
[5]Integrated Climate Data Center (ICDC), Center for Earth System Research and Sustainability (CEN), University of Hamburg, Hamburg, Germany
[6]Université de Toulouse, LEGOS (CNES/CNRS/IRD/UT3), Toulouse, France
[7]Norwegian Polar Institute (NPI), Tromsø, Norway
[8]Arctic Geophysics, University Centre in Svalbard (UNIS), Longyearbyen, Svalbard, Norway
[9]National Center for Climate Research (NCKF), Danish Meteorological Institute, Copenhagen, 2100, Denmark
[*]These authors contributed equally to this work.

**Correspondence:** Ida Birgitte Lundtorp Olsen (ilo@dmi.dk)

**Abstract.**

Sea ice altimetry currently remains the primary method for estimating sea ice thickness from space, however time-series of sea ice thickness estimates are of limited use without having been quality-controlled against reference measurements. Such reference observations for sea ice thickness validation in the polar regions are sparse and rarely presented in a format matching the satellite-derived products. Here, the first published comprehensive collection of sea ice reference observations including freeboard, thickness, draft and snow depth from sea ice-covered regions in the Northern Hemisphere (NH) and the Southern Hemisphere (SH) is presented. The observations have been collected using airborne sensors, autonomous drifting buoys, moored and submarine-mounted upward-looking sonars, and visual observations. The data package has been prepared to match the spatial (25 km for NH and 50 km for SH) and temporal (monthly) resolutions of conventional satellite altimetry-derived sea ice thickness data products for a direct evaluation of these. This data package, also known as the Climate Change Initiative (CCI) sea ice thickness (SIT) Round Robin Data Package (RRDP) was produced within the ESA CCI sea ice project. The current version of the CCI SIT RRDP covers the polar satellite altimetry era (1993–2021) and is part of ongoing efforts to keep the dataset updated. The CCI SIT RRDP has been collocated to satellite-derived sea ice thickness products from CryoSat-2, Envisat and ERS-1/2 produced within ESA CCI and the Fundamental Data Records for Altimetry (FDR4ALT) project to demonstrate the overlap and inter-comparison between the reference observations and satellite-derived products. Here, the



CCI SIT RRDP is introduced along with examples of its use as a validation source for satellite altimetry products, where the averaging, collocation and uncertainty methodology is presented and their advantages and limitations are discussed.

# 1 Introduction

Satellite altimetry is often used to derive sea ice thickness at reasonable temporal and spatial resolutions by detecting and converting freeboard i.e., the height of the sea ice above the local sea level depending on scattering horizon, into thickness assuming the ice to be in hydrostatic equilibrium (e.g., Laxon et al., 2003, 2013; Ricker et al., 2014; Kacimi and Kwok, 2020; Fons et al., 2023).

Typically, either radar (Ku-band) or laser altimetry is used to obtain derived freeboards and thicknesses. This is achieved based on the assumption that a laser measures the total freeboard (snow + sea ice freeboard), whereas for satellite-derived Ku-band radar observations it is commonly assumed to measure the sea ice freeboard once a correction of the slow-down of the radar wave propagation in snow ($c_s$) (Ulaby et al., 1986) has been applied. Without such correction, the freeboard measured by the Ku-band radar is commonly referenced as the radar freeboard. The assumption of full penetration is based on laboratory experiments over cold and dry snow conditions Beaven et al. (1995). To estimate the correction for the slow-down of the radar wave propagation in snow, a measure of snow depth and snow density is necessary (Mallett et al., 2020). The radar freeboard will in principle be located below the snow-ice surface as illustrated in Fig. 1. However, this is not a physical signal, but simply caused by the fact that the speed of light in air ($c$) is larger than the speed of light in snow ($c_s$), causing the conversion of travel-time to range (when not accounting for slow-down) being longer than reality. The height of the radar freeboard also includes other factors that impacts the scattering horizon and thereby the derived freeboard, such as e.g., roughness (Landy et al., 2020), basal snow salinity (Nandan et al., 2017), flooding (King et al., 2018; Kacimi and Kwok, 2020; Fons et al., 2023) and in-complete penetration (e.g. Willatt et al., 2010, 2011; Ricker et al., 2014; Stroeve et al., 2020b). These factors prevents the radar signal from penetrating all the way to the snow-ice surface.

The different freeboards (total or sea ice freeboard) can be converted into an estimate of sea ice thickness assuming that sea ice is in hydrostatic equilibrium and additional information on the snow depth and the densities of snow, sea ice and water is available (Wadhams et al., 1992). From these parameters, the snow depth and sea ice densities comprise the largest uncertainties in the sea ice freeboard to thickness conversion (e.g. Giles et al., 2007; Zygmuntowska et al., 2014; Ricker et al., 2014; Kern et al., 2015) with a higher sensitivity to snow depth errors in the total freeboard to sea ice freeboard conversion (Giles et al., 2007). While the impact of snow mass (snow depth and density) error decreases with increasing total freeboard due to a lower snow to ice mass ratio, the impact of sea ice density scales with sea ice freeboard. Thus, for thinner sea ice the snow load error may be more relevant for the sea ice thickness error budget, while for thicker sea ice errors in sea ice density may be the largest retrieval error component. The point where sea ice density error contribution exceeds the snow mass error contribution depends on the actual sea ice and snow conditions.

The snow depth uncertainties arises primarily due to the fact that there is currently no consistent snow depth dataset at the same temporal and spatial resolution as the satellite altimetry observations covering the entire polar satellite altimetry era





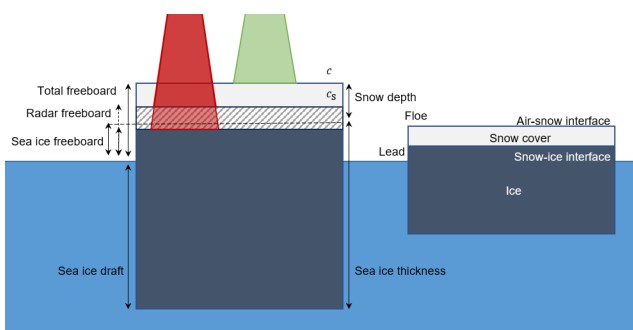

**Figure 1.** Schematic of the different sea-ice-altimetry-related terms. Radar (Ku-band) and laser altimetry observations and their expected penetration into the snow pack are shown by the red and green beams, respectively. Not to scale. The shaded area denote the uncertainty related to penetration of radar and slow-down of propagation speed, which is depending on the snow conditions, and impacts (along with other things) the retrieved radar freeboard.

(1993–present). The Arctic available monthly climatologies (e.g. Warren et al., 1999; Shalina and Sandven, 2018) or modified versions (Kurtz and Farrell, 2011) based on field observations often lack information of inter-annual variations and trends, and various snow depth products from models or temporally limited multi-sensor observations have shown large discrepancies (Zhou et al., 2021), thus there is limited consensus between the available snow depth estimates. In the Antarctic, one must currently rely on snow depths derived from e.g. passive microwave sensors (e.g. Markus and Cavalieri, 1998; Comiso et al., 2003; Shen et al., 2022), where the methods are most reliant over first-year ice, which is the predominant ice type. The record dates back to the early 1990s and thus covers the entire polar satellite era. However, the method is challenged by the fact that it cannot reliably measure snow depths exceeding 50 cm, which are common in the Antarctic (Lawrence et al., 2024). Additionally, numerous publications (e.g. Worby et al., 2008a; Markus et al., 2011; Ozsoy-Cicek et al., 2011) have demonstrated that the method underestimates snow depth over deformed sea ice by a factor of 2 or even 3. Snow depth models are lacking in the Antarctic, but first efforts have recently been published by Lawrence et al. (2024). Dual-frequency satellite altimetry derived snow depth estimates for both Arctic (Guerreiro et al., 2016; Lawrence et al., 2018; Kwok et al., 2020; Garnier et al., 2021; Kacimi and Kwok, 2022; Fredensborg Hansen et al., 2024) and Antarctic (Kacimi and Kwok, 2020; Garnier et al., 2021) are promising, but limited to the period when the NASA ICESat mission was operational (2003–2008) and after SARAL/AltiKa was launched (in 2013) and thus, do not cover the entire polar satellite era dating back to 1993. In addition, efforts are being made to better understand the penetration capabilities of Ku-band throughout the seasons and how that impacts the signal in preparation for the upcoming dual-frequency radar altimetry mission, CRISTAL (Copernicus Polar Ice and Snow Topography Altimeter, see Kern et al., 2020).

Conventional methods for estimating Arctic sea ice thicknesses from satellite altimetry use sea ice densities depending on sea ice types i.e., first-year ice (FYI) and multi-year ice (MYI) following Alexandrov et al. (2010). According to Jutila et al. (2022), these sea ice densities might not represent the present state of the sea ice as their results show higher average bulk densities for both ice types. The dual-density approach also introduce auxiliary sea ice types to be included in the sea ice





freeboard to thickness conversion, which in case of miss-classification may introduce relatively large errors (e.g. Haas et al., 2017; Kern et al., 2015). For the Antarctic sea ice, either a singular value for sea ice density assuming full coverage of FYI is used (Kacimi and Kwok, 2020) or, alternatively, the sea ice freeboard to thickness conversion takes into account a single combined density for snow and ice to accommodate for the special sea ice conditions in the Antarctic with negative freeboards

and flooded ice (e.g. Kern et al., 2016; King et al., 2018). More recent attempts by Fons et al. (2023) uses seasonal variable ice densities, however, most methods rely on in situ measurements collected by Worby et al. (2008a) and Buynitskiy (1967).

With altimetry observations ranging back to the beginning of 1990, there is a potential for developing long time-series of sea ice freeboard and thickness for climate analyses (Quartly et al., 2019). The European Space Agency (ESA) Climate Change Initiative (CCI) for sea ice (https://climate.esa.int/en/projects/sea-ice/, last access: 28 February 2024) has the ultimate

objective of producing Climate Data Records (CDR) of the Essential Climate Variable (ECV) sea ice thickness (SIT) (and sea ice concentration) from various altimetry (and passive-microwave-radiometry) missions without inter-satellite-mission biases. This presents challenges due to the differences in coverage, resolution and instrument design between the missions. The CCI SIT project utilizes observations from the European Remote Sensing Satellites-1 and -2 (ERS-1/2, ESA, 2013), the Environmental Satellite (Envisat, Dubock et al., 2001) and CryoSat-2 (Wingham et al., 2006). ERS-1 launched in 1991 (in

orbit until 2000) and its follow-up mission, ERS-2, launched in 1995 (in orbit until 2011), were equipped with, amongst others, the Radar Altimeter (RA) instrument. RA was a pulse-limited Ku-band radar altimeter (13.8 GHz), that acquired data in two operating modes: ocean or ice, and had a footprint diameter of 16-20 km (depending on sea state). We note, that RA on ERS-1 was only active for specific sea ice or geodetic campaigns from 1993–1996 and thus, the satellite altimetry time-series currently only range back to 1993. To continue data acquisition, Envisat was launched in 2002 (in orbit until 2012) which carried RA-

2, a dual-frequency pulse-limited radar altimeter operating at Ku-band (13.575 GHz) and S-band (3.2 GHz) with a footprint diameter of 20 km for Ku-band (Roca et al., 2009). ERS-1/2 and Envisat flew in sun-synchronous polar orbits and acquired data up to 81.5°N/S latitude. In 2010, the first-ever synthetic aperture radar (SAR) interferometric radar altimeter (SIRAL) was launched on the CryoSat-2 mission (Wingham et al., 2006). CryoSat-2's SIRAL measures in Ku-band (13.575 GHz), reaches 88° N/S latitude and has an improved spatial resolution with a footprint size of about ∼300 m along-track and ∼1.65 km

across-track (e.g. European Space Agency, 2019) when operating in SAR, which is the main mode used over sea ice, and provides observations acquired in interferometric SAR (SARIn) mode for coastal areas or over ice margins. The improved spatial resolution of CryoSat-2 when operating in SAR mode paved the way for more recently launched satellites (Sentinel-3A/B and upcoming C/D, and Sentinel-6 Michael Freilich), which all use this technique. ERS-1/2, Envisat and CryoSat-2 have together covered a period of 30 years, and therefore present an opportunity of creating consistent and quality-controlled

time-series of sea ice freeboards and thicknesses.

During the first phase of the CCI sea ice project, time-series of consistent sea ice freeboards and thicknesses were produced for the period of 2002–2021 using observations from Envisat and CryoSat-2 (Kern et al., 2015; Schwegmann et al., 2016; Paul et al., 2018) for both hemispheres. These are herein referred to as the CCI SIT CDR. Stand-alone CDRs are of little use unless the products have been quality-controlled against relevant reference measurements for consistency. Evaluation of the

satellite-derived sea ice freeboards and thicknesses is a crucial part of the CCI SIT project. For a complete validation of the





satellite-derived sea ice freeboards and thicknesses we need coincident observations of sea ice freeboards, sea ice thickness, snow depth and densities of snow, ice and water with the same temporal and spatial scales covering the polar satellite altimetry era. However, reference measurements are sparsely and unevenly distributed in the Arctic, with even less availability in the Antarctic, due to the harsh environment and cost-expensive access and often only measure one of the variables e.g., freeboard,
total thickness, draft or snow depth.

In this manuscript, we present a collection of publicly available sea ice reference measurements (to our knowledge the first set tailored for satellite data validation) relevant for sea ice thickness evaluation; including primary variables of freeboard, thickness, draft and snow depth acquired during the period 1993–2021 from different sources. Upon availability, surface and air temperature are provided as secondary variables. The collection of reference measurements is herein referred to as the
CCI SIT RRDP. In the CCI SIT RRDP, the reference measurements have been gridded to align with the scales and resolution of commonly provided satellite-derived sea ice thickness and freeboard-products (25 km in the NH and 50 km in the SH), and collocated with CCI SIT CDR version 3.0 (**http://cci.esa.int, last accessed on April 26, 2024**) including time-series from CryoSat-2 and Envisat. We have further collocated the CCI SIT RRDP with time-series of ERS-1/2 radar freeboards produced within the ESA Fundamental Data Records for Altimetry (FDR4ALT) project (Bocquet et al., 2023) to show the
data availability throughout the polar satellite altimetry era. The aim of CCI SIT RRDP is to provide a data package useful for evaluating satellite-derived sea ice thickness products by collocating the sea ice freeboards, thicknesses and derived drafts from the CCI SIT CDR processors with the observations included in the CCI SIT RRDP. We further include reference observations of snow depth, and collocate to the auxiliary snow depths provided in the CDRs as the snow depths are used both in the radar-freeboard-to-sea-ice-freeboard and sea-ice-freeboard-to-thickness conversions, as already described. We do not include
sea ice, snow or water densities, as there are limited overlapping reference observations of these.

Since the reference observations are acquired by different instruments and methods, the uncertainties of the measurements are important to take into consideration. Some of the reference measurements are relatively old (before 2010), and have limited uncertainty information in the provided data product. Furthermore, some reference measurements (e.g., visual observations from ships) only include one constant uncertainty estimate per data product and not per measurement. Thus, determining the
uncertainty of each estimate in the CCI SIT RRDP is not a trivial task, but is nonetheless important and has been a significant part of producing this RRDP (see Sect. 4).

This paper presents the CCI SIT RRDP, explaining in detail the collocation and averaging methodology as well as the uncertainty characterisation. Furthermore, a comparison with satellite data is presented to demonstrate the usage and performance of the CCI SIT RRDP. Since the CCI SIT RRDP provides already pre-processed comparable data for evaluation of sea ice
altimetry-derived freeboards and thicknesses, as well as auxiliary snow depth estimates - for both hemispheres - this data package and the methodologies applied herein have the potential of becoming the reference for future comparisons of current and future SIT products. The collection of reference measurements is presented in Section 2. Access to the CCI SIT RRDP and related software code is provided in Section 8. Pre-processing steps, and the uncertainties of each data product and the applied uncertainty methodology are presented in Section 3 and Section 4, respectively. The satellite SIT CDRs are described
in Section 5.1, followed by a description of the comparability and collocation of the CCI SIT RRDP and satellite SIT CDRs





in Section 6. Results and discussions of the CCI SIT RRDP and inter-comparison with satellite-derived products are presented in Section 7 including the availability of the reference measurements and their advantages and limitations. Finally, Section 9 concludes the paper.

## 2 Description of RRDP reference measurements

The CCI SIT RRDP includes observations of freeboard (FRB), thickness (SIT), draft (SID) and snow depth (SD) in both the Arctic and Antarctic regions. Here, we use the term FRB as a general term for freeboard including total freeboard and sea ice freeboard as both are available in the CCI SIT RRDP. In addition, SIT includes total thickness (snow + sea ice thickness) and sea ice thickness. Some reference measurements provide additional information on surface temperature and air temperature. As the temperature has an impact on radar penetration depths (e.g. Giles and Hvidegaard, 2006; Willatt et al., 2011) we have

included the temperature observations in the CCI SIT RRDP. All available reference observations (from the included sources) for the polar regions for both hemispheres are part of the CCI SIT RRDP, providing us with observations throughout the entire year, although the coverage and availability are seasonally dependent, as can be seen in Fig. 5. Reference observations north of the satellite altimeter coverage i.e., the polar gap, has been included to prepare for evaluation of potential future satellite CDR gap filling products (e.g. by use of statistical methods such as kriging or optimal interpolation, see Gregory et al., 2021), but

also to be used as reference observations for models.

In total, data from 10 different sources in the Arctic and 4 different sources in the Antarctic are included in CCI SIT RRDP. The measurements are obtained from different platforms using varying methods i.e., airborne measurements, measurements from moorings and autonomous drifting buoys, ships, and submarines. These methods and data products will be described more thoroughly in the following sections. A complete overview of the data sources used in the CCI SIT RRDP for both

hemispheres is presented in Table 1 and links to the raw data are available from Table 2. Data sources are further illustrated in the Sankey diagram in Fig. 2, where the platform and methods of measurements are shown, along with the name of the data sources and the associated geophysical sea ice variables included in the CCI SIT RRDP. From here on, the data sources will be referred to by their abbreviations as provided in Table 1.

## 2.1 Airborne measurements

Measurements from both OIB and AEM-AWI are conducted from airborne platforms (see Fig. 2). These measurements provide a higher spatial resolution than satellite measurements, and a larger spatial coverage than in situ observations, however, they are usually temporally limited to a few days to weeks in specific months, primarily spring (March-April) in the NH and austral summer (October) in the SH, depending on when and whether an airborne campaign was conducted (see Fig. 5).





**Table 1.** Overview of reference measurements and their sources included in the CCI SIT RRDP for both the Northern (NH) and Southern (SH) hemispheres.

| Campaign name or responsible | Description | Location | Abbreviation |
|---|---|---|---|
| *Northern Hemisphere (NH)* | | | |
| North Pole Environmental Observatory | Stationary moored upward-looking sonar | North Pole | NPEO |
| Fram Strait Arctic Outflow Observatory | Stationary moored upward-looking sonar | Fram Strait | NPI-FS |
| Beaufort Gyre Exploration Project | Four stationary moored upward-looking sonars | Beaufort Sea | BGEP |
| Russian-German TRANSDRIFT project | Four stationary moored upward-looking sonar | Laptev Sea | TRANSDRIFT |
| Arctic Shipborne Sea Ice Standardization Tool | Visual observations from ice breakers | Arctic Ocean | ASSIST |
| Submarine Arctic Science Program | Submarine-mounted upward-looking sonar | Arctic Ocean | SCICEX |
| Cold Regions Research and Engineering Laboratory | Ice mass balance buoys | Arctic Ocean | IMB-CRREL |
| Alfred Wegener Institute | Snow depth buoys | Arctic Ocean | SB-AWI |
| Alfred Wegener Institute | Airborne electromagnetic measurements | Lincoln Sea/Beaufort Sea | AEM-AWI |
| NASA's Operation IceBridge | Airborne laser and radar altimetry and snow radar | Lincoln Sea/Beaufort Sea | OIB |
| *Southern Hemisphere (SH)* | | | |
| Alfred Wegener Institute | Stationary moored upward-looking sonar | Weddell Sea | AWI-ULS |
| NASA's Operation Ice Bridge | Airborne laser and radar altimetry | Weddell Sea | OIB-SH |
| Alfred Wegener Institute | Snow depth buoys | Weddell Sea | SB-AWI-SH |
| Antarctic Sea ice Processes and Climate | Visual observations from ice breakers | Southern Ocean | ASPeCt |



**Table 2.** Direct data access is provided to all included reference data by using the DOI/url links. DOI's are used when available.

| Campaign | Reference | DOI/URL | Additional notes |
|---|---|---|---|
| OIB (NH, SH) | Kurtz et al. (2016), Kurtz et al. (2015) | https://doi.org/10.5067/GRIXZ91DE0L9 | QuickLooks (QLs) |
| | | https://doi.org/10.5067/G519SHCKWQV6 | IDSC4. SH data is available from Operation IceBridge Data Portal, similar to IDCS4 |
| AEM-AWI | Grosfeld et al. (2016) | https://data.meereisportal.de/relaunch/airborne?lang=de | Data from 2012-2019 is not publicly available yet |
| ASSIST | ASSIST (2006) | https://icewatch.met.no/ | Data is repeatedly added, hence, providing more data than what is used in this study |
| ASPeCt | Worby et al. (2008b) | https://aspect.antarctica.gov.au/data.html | |
| | Kern (2020) | https://doi.org/10.26050/WDCC/ESACCIPSMVSBSIOV2 | |
| IMB-CRREL | Perovich et al. (2022) | http://imb-crrel-dartmouth.org/archived-data/ | |
| SB-AWI (NH & SH) | Nicolaus et al. (2017) | https://doi.org/10.2312/polfor.2016.011 | Data available by: Maps & Data -> Method -> Autonomous measurements -> Snowbuoy |
| BGEP | BGEP (2003) | https://www2.whoi.edu/site/beaufortgyre/data/mooring-data/ | |
| AWI-ULS | Behrendt et al. (2013b) | https://doi.org/10.1594/PANGAEA.785565 | |
| TRANSDRIFT | Belter et al. (2019), Belter et al. (2020) | https://doi.pangaea.de/10.1594/PANGAEA.899275 | |
| | | https://doi.pangaea.de/10.1594/PANGAEA.912927 | |
| NPI-FS | Sumata et al. (2021) | https://doi.org/10.21334/npolar.2021.5b717274 | |
| NPEO | Morison et al. (2016) | https://doi.org/10.5065/D6P84921 | |
| SCICEX | NSIDC (1998) | https://doi.org/10.7265/N5930R3Z | From 1993-2014 |
| | SCICEX (2009, 2014) | https://doi.org/10.7265/N54Q7RWK | From 1960-2005 |
| ERS-1 & ERS2 | (Bocquet, 2023) | https://doi.org/10.6096/ctoh_sit_2023_01 | |
| CryoSat-2 | (Hendricks, 2024c), (Hendricks, 2024a) | https://catalogue.ceda.ac.uk/uuid/c6504378f78c4ecd9f839b0434023eff | Northern Hemisphere |
| | | https://catalogue.ceda.ac.uk/uuid/861ad3c7f3a34ebd8be6f618a92bd8e3 | Southern Hemisphere |
| Envisat | (Hendricks, 2024b), (Hendricks, 2024d) | https://catalogue.ceda.ac.uk/uuid/92eb2ba942074bec804af6a8b5436bee | Northern Hemisphere |
| | | https://catalogue.ceda.ac.uk/uuid/af96a1ec493f49caa39dc912d15f2b17 | Southern Hemisphere |







**Figure 2.** Sankey diagram providing an overview of the data sources used in the CCI SIT RRDP and how they are acquired, by whom/when and what has been observed. The diagram is shown by four categories; sensor or measurement type, platform, campaigns (Table 2), derived geophysical variable, and their dependencies. We note that the data volume included in the CCI SIT RRDP is represented here by numbers (in scientific notation) and percentages, and it is dependent on the processing level. Platform data volume denotes raw data, while campaign/geophysical variables refer to processed data. Note that the dominant data sources (at different processing levels) are highlighted in bold. In particular, the platform/mooring contributes more than 80% of the data, which primarily reflects the large time-series, high sampling frequency, and temporal coverage of the buoys. In contrast, campaign (OIB-NH, ASPeCT, or SCICEX) primarily reflect spatial coverage, since these numbers refer to the processed CCI SIT RRDP. In total, SIT and SD account for approximately 60% of the CCI SIT RRDP. The notation O(number) denotes an approximation.



## 2.2 Campaign details: OIB and AEM-AWI

OIB's primary objective was to bridge the gap between NASA's ICESat (2003-2009) and ICESat-2 (2018-onwards) satellite missions. Over 12 different aircraft types was used during its duration from 2009-2019 and within this time span, several updates to the instruments used for surveying were made. For a detailed overview of instruments used for different periods, along with an assessment of the impact on the obtained data, see MacGregor et al. (2021). The OIB reference measurements used in this study consist of data from the IceBridge "L4 Sea Ice Freeboard, Snow Depth, and Thickness", Version 001 (IDCS4), data product in the period of 2009–2013. Data subsequent to this are provided as quick-looks (Kurtz et al., 2016), which means that significantly less processing has been done. Quick-looks are processed from 2012–2019, but are only used from 2014–2019 due to IDCS4 being available until 2014. The measurands that we compare to satellite observations are the sea ice FRB (total FRB subtracted the snow depth) and SD as well SIT that is derived from laser and radar observations with additional sea ice density parametrization. The officially published Antarctic campaign data is limited to total FRBs from the airborne topographic mapper (ATM) for the 2009 and 2010 campaigns with no snow depth estimates provided and hence, no sea ice thickness estimates. To produce the OIB product, post-processing algorithms have been used (Kurtz et al., 2013), which are based on observations from the ATM laser altimeter system, a digital camera (Section 2.2.1) and a snow radar (Section 2.2.2).

AEM-AWI measurements are conducted using an electromagnetic (EM) sounding device (known as the "EM-Bird", see Section 2.2.3) dedicated to measuring the total thickness. In total, 25 campaigns are included in AEM-AWI dataset as provided in Olsen and Skourup (2024a) along with information on the platform and measurement type for each campaign. AEM-AWI has data from 2001–2019 but has not been measured consistently each year.

### 2.2.1 Airborne topographic mapper (ATM) and digital camera

The main components of ATM are two conically scanning laser altimeters that measure the surface elevation along the path of the aircraft at 15° and 2.5° off-nadir angle, respectively (MacGregor et al., 2021). The ATM measures surface elevation with respect to the WGS-84 reference ellipsoid by incorporating measurements from global navigation satellite system (GNSS) receivers and inertial navigation system attitude sensors. Measurements from ATM are subsequently converted to measurements of total freeboard, by subtracting the instantaneous sea surface height (the local sea level obtained from lead measurements) from the measured elevation height. Determination of the sea surface height involves corrections of geoid height, tides, atmospheric pressure and the dynamic sea surface e.g. waves. During these procedures, information from the geo-referenced images from the digital cameras is used to support the identification of leads, which are used as tie-points for the instantaneous sea surface height. For more information about the ATM and the subsequent processing, see Kurtz et al. (2013).

### 2.2.2 Snow radar

SD is recorded using an ultra-wide frequency-modulated-continuous-wave (FMCW) radar at either S/C (2-8 GHz) or S/Ku (2-18 GHz) band (MacGregor et al., 2021). The snow radar measures the return radar signal as a function of time, which is scattered from the illuminated area below the aircraft. SD is determined by identifying the air-snow and snow-ice interfaces





(see Fig. 1 for definition of the interfaces) in the received signal and converting the time difference between these interfaces to SD accounting for slow-down of propagation speed using the refractive index of snow (Kurtz et al., 2013).

### 2.2.3 EM-Bird

The EM-bird senses the distance of the sensor to the ice-water interface using frequency-domain EM induction sounding capitalizing on the substantial difference of electrical conductivity between the sea ice and snow layers compared to the ocean (Haas et al., 2009). Subtracting the instrument distance to the air-snow surface, measured by an integrated laser, from the distance to the ice-water surface yields the total thickness. The EM probe is towed by a helicopter or aircraft approximately 10-20 m above the surface of the sea ice. The temporal extent of available relevant total thickness data in the AEM-AWI dataset

is 2001, 2003-2017 and 2019. Measurements of the total freeboard are provided for 2004 (IRIS, GreenIce) and 2007 (PolICE) campaigns in the AEM-AWI dataset (Olsen and Skourup, 2024a). These were derived from the helicopter EM-integrated laser.

### 2.3 Stationary moorings

In the CCI SIT RRDP, we have included data from stationary moorings equipped with upward looking sonar (ULS). The observations are the only sea ice reference measurements, which are fixed to a specific geographic location, and provide

continuous measurements throughout the year. Even though the individual measurements are point measurements, the sea ice drift causes time-averaged sea ice drafts from ULS' to provide information of the sea ice pack representative of a larger area. Many of the existing ULS' provide long time-series (Fig. 4), and thus provide reference measurements ideal for evaluation of long-term multi-satellite climate data records. An ULS is an instrument targeted at measuring the sea ice draft (the submerged part of the ice). The instrument emits sound pulses, and detects their echo return after being reflected on the bottom of the ice

or the water level in between the ice floes. From these observations, the sea ice draft is derived after applied corrections. For basic principles see Melling et al. (1995).

In the Arctic, we include four sources providing stationary ULS data; the North Pole Environmental Observatory (NPEO) located at the North Pole with data from 2001–2010, the Fram Strait Arctic Outflow Observatory of the Norwegian Polar Institute (NPI-FS) located in the Fram Strait with data from 1990–2018, the Beaufort Gyre Exploration Project (BGEP) in

the Beaufort Sea providing data from 2003–2018, and the Russian-German TRANSDRIFT project (TRANSDRIFT) in the Lincoln Sea and Beaufort Sea with data from 2003 to 2016. Data from the TRANSDRIFT project consist of ULS data from 2013–2015 and Upward-Looking acoustic Doppler current profiler (ADCP) from 2003–2016 (Belter et al., 2020).

In the Antarctic, there are ULS draft observations from AWI (AWI-ULS) moorings in the Weddell Sea, where data was collected from 1990 to 2011. We note, that currently there are several ULS' stationed around the Arctic and Antarctic ensuring

the continuation of the mooring time-series of ice draft, however data is not available in near-real-time due to the sensor being submerged under water. Routine deployment and collection efforts are needed to deploy and retrieve ULS data, which is then processed afterwards. Hence, the lag-time for data collection is significant compared to other data sources e.g., autonomous drifting buoys with Iridium link.





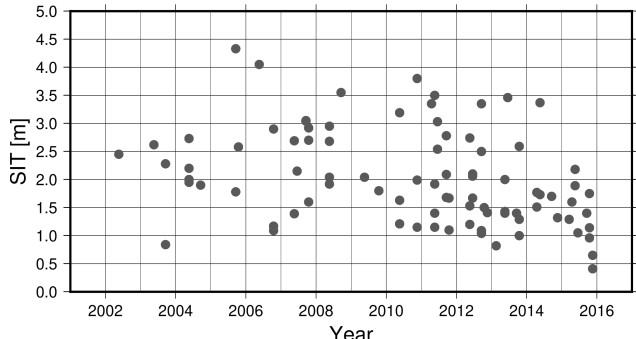

**Figure 3.** Initial thickness for the 87 IMB-CRREL buoys included in this study.

## 2.4 Drifting buoys

Drifting buoys are autonomous systems, installed on selected sea ice floes, that drift freely with the ice compared to being anchored to one location as moorings. Depending on the buoy model, drifting buoys provide, along with other variables, the evolution of the thermodynamic growth and melt of the ice and/or changes in the snow depth at the specific point where they are deployed, and representativeness of the data acquired by these buoys depend on the initial snow and ice conditions at the time of deployment.

The derived information from drifting buoys are limited to point measurements, and do not capture ice distribution on larger spatial scales nor the changes in the ice thickness distribution due to deformation. In the CCI SIT RRDP, we have included data from ice mass balance (IMB) buoys with data from 2003–2015, and from dedicated acoustic snow depth buoys (SB) with data from 2013–2020. SB also has data available for the southern hemisphere from 2013–2023.

     IMB buoys measure thermodynamic contribution to changes in the mass balance of the sea ice. The main component is
thermistor strings mounted vertically throughout the snow and ice column measuring the temperature profile, along with acoustic sounders placed above and below the ice, measuring the top of the snow/ice and the bottom of the ice, respectively. In addition, the buoys are typically equipped with a barometer and an air temperature sensor. The IMB buoy data in the CCI SIT RRDP is obtained from the Cold Regions Research and Engineering Laboratory (IMB-CRREL) (Perovich et al., 2022). From the IMB-CRREL buoys, we include measurements of SIT and SD, along with surface and air temperature. Typically,
3–6 IMB buoys are deployed each year in the Arctic Ocean, with the most regular deployments focused in the Beaufort Sea and at the North Pole with a typical survival period of 1–2 seasons. The time interval between subsequent mass balance data measurements (SD and SIT) varies for different buoys and depend on the buoy model and the year of deployment. In general, the mass balance data is measured approximately every four hours for the majority of buoys, but several buoys provide measurements every two hours and some (2002A, 2003A) only twice a day, while others (2015I, 2015J, 2015K) have
measurements every hour. As such, the IMB buoys provide spatially local measurements, due to their fixed location on an ice floe, but with a relatively high temporal resolution.





The IMB-CRREL buoys initial thicknesses i.e., the ice thickness at deployment, are shown in Fig. 3. Of the 87 buoys included here, only 1 buoy (2015H) has an initial SIT<0.5 m. In general, they tend to be deployed in ice thicker than 1 m with few exceptions (2003C, 2013A, 2015H, 2015I, 2015K) and most of them (21 out of 29 buoys) in the perennial sea ice cover (MYI) with initial thicknesses >2 m prior to 2009 to decrease the likelihood of damage to the buoy due to e.g., sea ice deformation events and thereby prolonging its potential life span. Buoys deployed in MYI do not capture the seasonal cycle (Polashenski et al., 2011). Post-2009, more buoys are deployed in ice with an initial thickness <2 m (39 out of 58) with a minimum initial thickness of 0.41 m (2015H), i.e., in the seasonal ice cover (FYI). This is consistent with the design of the first IMBs to be adapted and well-suited for deployments in MYI (Richter-Menge et al., 2006b). An optimized buoy design to better fit deployments in the seasonal ice zones was first tested in 2009 according to Polashenski et al. (2011).

Snow depth buoys (SB) measure relative changes in snow height, that is the accumulation of snow since deployment. These are then calibrated against the initial snow depths measured during deployment in order to retrieve the absolute snow depth values. In the Alfred Wegener Institute snow depth buoys (SB-AWI) (Nicolaus et al., 2017), the measurements are made with four ultrasonic snow depth sensors that are installed on a mast-attached platform. SB-AWI measurements are available for both hemispheres. The data transmission interval for SB-AWI is approximately once per hour resulting in similar spatial and temporal characteristics to IMB-CRREL.

## 2.5 Ships

Collected and archived ship-based observations are provided via the Ice Watch program (Hutchings et al., 2018) for the NH. These are visual observations, that are recorded with the ASSIST (Arctic Shipborne Sea Ice Standardization Tool) following an established Ice Watch protocol. Reported observations may include e.g. sea ice concentration and thickness, stage of growth or melt, state of the snow cover and surface roughness, and may be inconsistent in which variables are recorded within a particular observation and at what quality depending on the experience and qualifications of the observer (find a brief description of Ice Watch instructions for observers in Section 3.4). ASSIST data is available from 2006–2021 and contains data from 61 voyages. The Southern Hemisphere has a corresponding program called ASPeCt (Antarctic Sea Ice Process and Climate), which was established in 1997 by the Scientific Committee on Antarctic Research. Data until 2005 is available from the ASPeCt data archive and contains data from 83 voyages and 2 helicopter flights for the period 1980–2005. More recent additions (2002–2019) to the dataset have been processed and are publicly available (Kern, 2020). Links to data sources for ASPeCt and ASSIST are available from Table 2. Ship observations are, in a similar manner as the airborne campaigns, dedicated to individual cruises with a duration of 1–2 months. Many of the observations in SH are made from supply ships, which primarily take place during the Austral summers and tend to navigate thin or less consolidated ice. In general, ships, whenever possible, tend to avoid well consolidated and deformed ice to limit risks, which impacts the ice observations.

## 2.6 Submarines

The submarine dataset provide ULS SIDs, similarly to stationary moorings, except that the measurements are taken along trajectories and thus have a larger regional coverage. However, the data is collected only during dedicated cruises of 1–3





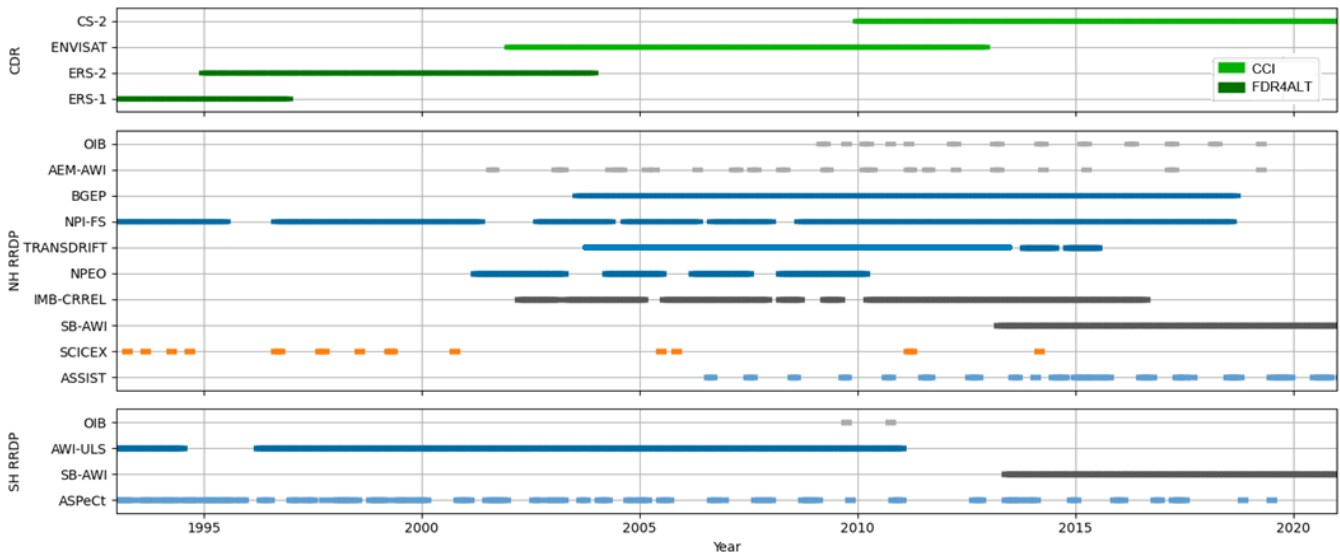

**Figure 4.** Timeline of all data included in the CCI SIT RRDP. The CDR observations are color-coded according to project as shown in the legend, whereas the colors of the RRDP reflect the type of reference observations as defined in Figure 5.

months duration. The submarine cruises are primarily military operations, which can imply that the data distribution to the common sea ice community can take several years due to restrictions on data sharing. The data included here were collected by the U.S. Navy and Royal Navy and are available for the Arctic Ocean. The temporal span of the data is from the 1st of February 1960 to the 30th of November 2005, along with data available in 2011 and 2014. Data from several other years (2012, 2016, 2018 and 2020) are currently being processed and evaluated for releasability.

## 3 Processing of RRDP reference measurements

Averaging of the reference data was performed using the Equal-Area Scalable Earth Grid in version 2 (EASE2) provided by the National Snow and Ice Data Center (NSIDC). EASE2 is based on a polar aspect spherical Lambert Azimuthal equal-area projection Brodzik et al. (2012) and the WGS-84 ellipsoid. The NH grid dimension is 5400 km x 5400 km with a spatial resolution of 25 km, resulting in a grid consisting of 432 x 432 grid-cells, whereas the SH grid has a spatial resolution of 50 km, resulting in a grid consisting of 216 x 216 grid-cells. The grid is centered on the geographic pole, meaning that the pole is located at the intersection of center cells. A temporal resolution of 30 days and a spatial resolution of 25 km for the Northern Hemisphere and 50 km for the Southern Hemisphere is used. Data obtained from stationary moorings, have only been temporally averaged, as these are fixed in space. The output data was subsequently temporally sorted and processed to fit into a standardized text format, as shown in Table 3. Since most campaigns only record some of the information in the standardized format, missing information was filled with NaN.





**Table 3.** Description of reference data structure in CCI SIT RRDP.

**SIT, SD and FRB reference data files**

| | | | | | | | |
|---|---|---|---|---|---|---|---|
| obsID | Observation Identifier | date | Median of dates in gridcell | lat | Latitude of grid center | lon | Longitude of grid center |
| SD | Average SD in gridcell | SDstd | Standard deviation of SD in gridcell | SDn | Number of SDs in average | SDunc | Uncertainty of gridcell SD (*) |
| SIT | Average SIT in gridcell | SITstd | Standard deviation of SIT in gridcell | SITn | Number of SITs in average | SITunc | Uncertainty of gridcell SIT (*) |
| FRB | Average FRB in gridcell | FRBstd | Standard deviation of FRB in gridcell | FRBn | Number of FRBs in average | FRBunc | Uncertainty of gridcell FRB (*) |
| Tsur | Average Surface temperature | Tair | Average Air temperature | wSD | Warren Snow Depth (**) | w-rho | Warren Snow Density (**) |
| pp-flag | Pre-processing flag (*) | unc-flag | Uncertainty flag (*) | | | | |

**SID reference data files**

| | | | | | | | |
|---|---|---|---|---|---|---|---|
| obsID | Same as above | date | Same as above | lat | Same as above | lon | Same as above |
| SID | Average SID in gridcell | SIDstd | Standard deviation of SID in gridcell | SIDn | Number of SIDs in average | SIDunc | Uncertainty of gridcell SID (*) |
| wSD | Same as above | w-rho | Same as above | pp-flag | Same as above | unc-flag | Same as above |

(*) see sec. 4 for a description of how uncertainties are determined for each campaign (**) from Warren et al. (1999)




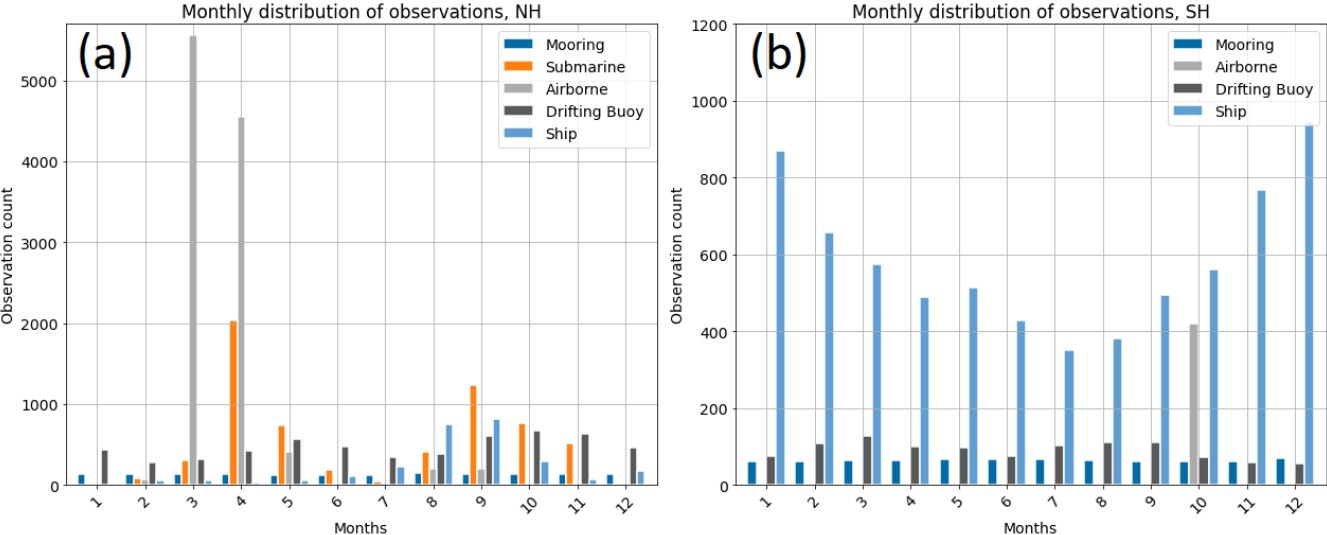

**Figure 5.** Seasonal distribution of reference data in CCI SIT RRDP; (a) NH and (b) SH categorised based on sensor type.

All of the collected reference data underwent some degree of processing procedures prior to using Ease-Grid 2.0. In most cases, the necessary pre-processing steps involved converting the date and coordinate formats to the desired output format and/or automatically extracting location and/or date information from the file header. These pre-processing steps are considered standard procedures. In addition, several reference data subsets needed additional pre-processing steps due to e.g., incomplete information of time and/or position. As a result, a pre-processing (pp) flag is included in the final data file, indicating if the file required supplementary pre-processing steps, which could be associated with higher uncertainties. As the required level of additional pre-processing varies, the pre-processing flag is categorized into the following:

  0. : No additional pre-processing performed

  1. : Very minor pre-processing

  2. : Minor pre-processing

  3. : Major pre-processing

The subsequent sections describe required pre-processing for campaigns/data with non-zero pre-processing flags.

### 3.1 Pre-processing of AEM-AWI

Several files from AEM-AWI required additional pre-processing due to missing time information, see data overview in Olsen and Skourup (2024a) for a file-wise overview. For files with missing time information, the output date-time is given as the date plus an arbitrary time stamp of 00:00:00 of the respective day. These files are marked by a pp-flag of category 1. Files within AEM-AWI either contain measurements of total FRB or total thickness. It was therefore decided to split the final product into



two files, one with all the measurements of total ice thickness and one with all measurements of total freeboard (see appendix in Olsen and Skourup (2024a) for a filewise overview).

## 3.2 Pre-processing of NPI

Stationary mooring data from NPI is given a pp-flag of category 1, due to the raw data already being processed into monthly means by the data source (see Table 4). This means that no standard deviation or information about the number of measurements used for each average value is available from the dataset. However, an updated version of the data (Sumata, 2022) contains an estimate of the number of samples per observation given as in the order of $10^4$ for data obtained with the ES300 instruments (until Sep 2006) and in the order of $10^6$ for data obtained with the IPS4/5 instruments. This information has been added to the CCI SIT RRDP, but users should be aware that the number is an approximation.

## 3.3 Pre-processing of IMB-CRREL

IMB-CRREL data is structured so that it contains a mass balance file, a position file, a file with meteorological data and a file with ice temperature data. SD and SIT information is available from the mass balance file, coordinates from the position file, air temperature from the meteorological data file and ice surface temperature from the temperature data file. However, these files have different date arrays and do therefore not generally coincide in time, signifying that they cannot be automatically combined. To combine information from these files, a starting point is taken in the mass balance data, as this is the most important data for this study. Therefore, data from the other files are co-located to this by assigning data with the smallest time difference. As data does not overlap perfectly in time, a pp-flag of category 1 is given as default. If the time difference between the position data and the mass balance data is more than 12 hours for any value used in a grid point, then this grid point is given a pp-flag of category 2. IMB SD data is recorded as a change in snow depth compared to an initial measurement. However, some data lacks records of initial snow depth. This data is given a pp-flag of category 3 and should be used with care.

## 3.4 Pre-processing of ship measurements

The IceWatch manual provides guidelines for how to observe SIT and SD from the ships. The standard procedure for an ice watch is to make observations every hour while the ship is in motion. The ice is observed, depending on actual visibility conditions, up to a mile from the ship during a 10-minute observation period, which is usually performed on the bridge of the ship. The aim is to classify up to three prevalent ice types/classes that combined cover the most area. Of these, the thickest is the *primary* ice type, and the thinnest is the *tertiary* ice type. The area of *primary*, *secondary* and *tertiary* ice should sum to the total ice concentration (Hutchings et al., 2018).

Due to this acquisition method, data from ASSIST and ASPeCt must undergo pre-processing to combine the observations of the individual ice types into one. The thickness and partial concentration of each ice type are used to make a weighted average of the mean sea ice thickness within the observed area (1 nautical mile $\approx$ 1.85 km according to Hutchings et al. (2018)). The



following formulas show the computation of such averaged sea ice thickness estimates:

$$\text{SIT}_{\text{EP}} = \frac{C_P}{C_{\text{tot}}} \cdot \text{SIT}_P, \tag{1}$$


$$\text{SIT}_{\text{ES}} = \frac{C_S}{C_{\text{tot}}} \cdot \text{SIT}_S, \tag{2}$$

$$\text{SIT}_{\text{ET}} = \frac{C_T}{C_{\text{tot}}} \cdot \text{SIT}_T, \tag{3}$$

$$\text{SIT} = \text{SIT}_{\text{EP}} + \text{SIT}_{\text{ES}} + \text{SIT}_{\text{ET}} \tag{4}$$

Here, P, S and T stand for *primary*, *secondary* and *tertiary*, respectively. $C_{\text{tot}}$ is the total ice concentration, and $C_{\text{P, S, or T}}$ denote the ice concentration of the particular ice type. The combined SD is derived using the same weighting principle.

Reference data is used if the sum of the partial concentration adds up to the total concentration and at least one of the partial concentrations belonging to SIT/SD is defined (e.g. is not NaN). The calculation method only considers the average SIT/SD of
areas covered by sea ice. Thus, the estimates do not take into account whether the ship is primarily surrounded by water or ice. This can bias the estimates towards higher sea ice thickness and snow depth when compared to satellite measurements since it is not weighted by information of open water coverage. However, as will be shown and discussed in Section 7, comparisons between ship-based reference data and satellite products tend to show the opposite trend, with higher values of satellite products than reference data from ships. If the total sea ice concentration is recorded as 0, the corresponding SIT and SD are also set to
0. Due to these processing steps, all ASSIST and ASPeCt observations are marked by a pp-flag of category 2.

### 3.5   Pre-processing of submarine measurements

SCICEX submarine measurements are obtained over a long time span (1960-2014). During this time-period the information provided in the data acquisition files are not consistent and post-processing of different parts of the data have been treated by different institutions, which results in reference observation inconsistencies between different cruises. Nevertheless, as all
SCICEX data are subject to some level of interpolation, due to a lack of continuous measurements of time and position all the data is given pp-flags in categories 1 to 3.

Parts of the SCICEX data are known as the "analog subset" because it was derived from traces on paper rolls (SCICEX (2009, 2014), see General resources *Documentation for G01360 Analog Subset*). Each file in the analog subset contains drafts of one line segment and provides only the start and end coordinates, along with date information including the year, month and
the segment of the month in which measurements were obtained, given as the first, second or third part of the month. We are using the following date-time conversion for converting the segment of the month into a date containing day and time:





- 1st third = Day 5 at 00:00:00

- 2nd third = Day 15 at 00:00:00

- 3rd third = Day 26 at 00:00:00

Other files within SCICEX provide a specific day of the month and for these, we use the specified day and an arbitrary time at 00:00:00.

Spatial interpolation to obtain positions of each reference measurement is done using the inverse haversine formula from the Python package haversine 2.8.0 (released Feb 28 2023). Here the coordinates $(\phi, \lambda)$ are calculated iteratively by using the provided distances $(\delta d)$ between observations when available and the bearing $(\theta)$ between neighbouring points. When these

are not available, equal distance is assumed between subsequent measurements using the start and end positions.

## 4    Uncertainties of RRDP reference measurements

All reference data in the CCI SIT RRDP are related with some degree of uncertainty, however, except for OIB, they lack uncertainty information for individual data points. Instead, uncertainty quantification in the CCI SIT RRDP must rely on average errors, accuracies, or uncertainties reported in various studies. These sources are presented in Table 4 along with the

estimated uncertainty. It is important to note that the amount of uncertainty information varies greatly among the datasets, and several uncertainty estimates are based on assumptions (e.g., AEM-AWI), instrument accuracy alone (e.g., IMB-CRREL), or are only valid within a certain range. Therefore, an uncertainty flag is introduced to indicate situations where the uncertainty may be underestimated, overestimated, based on assumptions, or potentially flawed. Section 4.2.2 provides a description of the uncertainty for each reference dataset, including whether and why an uncertainty flag has been assigned. As the assumptions

linked to the uncertainty vary, the uncertainty flag is categorized as:

0.  No uncertainty assumption, individual uncertainties are provided for each measurement

1.  Uncertainty measures have some degree of distinction based on e.g. thickness, time of year or likewise

2.  The same uncertainty is assumed for all data

3.  The same uncertainty is assumed for all data and this uncertainty is linked to incorrect assumptions

Another concern is the interchangeable use of terms such as error, uncertainty, and accuracy, despite their distinct statistical meanings. Error represents the absolute deviation between measured and true values, accuracy describes a closeness of the agreement between measured and true values and is a qualitative term only, and uncertainty provides a quantification of the doubt of a measurement given as an estimate of the range within which the true value is expected to lie (Taylor, 1939; Bell, 1999). Therefore, while error is calculated based on a known true value, uncertainty is typically described by a confidence

interval or standard uncertainty within which the true value is expected to fall. Uncertainty is a measure of the random error in a sample, whereas systematic error is referred to as bias or an offset (Bell, 1999). Based on these definitions, when a paper refers to an error indicated by a $\pm$ value, it is here interpreted as uncertainty.





## 4.1 Uncertainty propagation in average calculation

The propagation of uncertainties in the final CCI SIT RRDP product is based on the principles outlined in Taylor (1939). According to Taylor's theorem, if the uncertainties of the measurements $x_1$ to $x_n$ are independent and random, then the uncertainty of the mean is obtained by summing the individual uncertainties in quadrature and dividing by the square root of the number of measurements (N).

$$\delta\bar{x} = \frac{1}{N}\sqrt{\delta x_1^2 + \delta x_2^2 + ... + \delta x_n^2}. \tag{5}$$

The upper bound of the uncertainty is the ordinary sum of the measurement uncertainties:

$$\delta\bar{x} \leq \frac{1}{N}(\delta x_1 + \delta x_2 + ... + \delta x_n) \tag{6}$$

In this study formula 5 is used for the propagation of uncertainties, hence, the uncertainties are assumed to be independent and random, as we do not have sufficient information to obtain full error covariance matrices. Nevertheless, this is not necessarily the case and it will result in some uncertainties being underestimated. The effect of this is largest for those campaigns where the number of observations per grid cell is large, which is particularly the case for the ULS observations, where several thousand measurements are averaged to obtain the final values in the CCI SIT RRDP. Therefore, we underline that it might be more appropriate to use the upper bound uncertainty in some cases, which is equivalent to the uncertainty estimates shown in Table 4. This is especially true for campaigns/data where we have the same uncertainty estimate for all input data, as is the case for the majority of the campaign data. Table 4 presents a summary of input uncertainty estimates for each campaign, along with a citation to the publication where the uncertainty estimate was originally sourced. In the following sections, we describe in more detail how the uncertainty estimates of each campaign are obtained and the underlying assumptions.

## 4.2 Original uncertainties of data

### 4.2.1 Airborne data

OIB data contain individual uncertainty estimates for FRB, SD and SIT measurements as the only data sources included in the CCI SIT RRDP. These uncertainties are based on variations in the sea ice properties, instrument, and inter-campaign algorithm changes. Therefore, OIB data is given an uncertainty flag of category 0. A detailed overview of how uncertainties are calculated is presented in Kurtz et al. (2013). The uncertainty of AEM-AWI depends on the SIT. For a single helicopter-borne EM measurement, the uncertainty is $\pm 0.1$ m over level ice. However, this uncertainty is inaccurate for deformed ice, as deformed ice thickness can be strongly underestimated due to the footprint of EM measurements over those 3D structures, and due to e.g., air pockets between the ice floes that have been pushed together (Haas et al., 2009; Mahoney et al., 2015). However, no uncertainty quantification for thicker sea ice has been made and therefore average sea ice thicknesses of more than 3 m are given an uncertainty flag of category 3, whereas sea ice thinner than 3 m are given an uncertainty flag of category

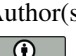



**Table 4.** Uncertainties related to the products in CCI SIT RRDP.

| Campaign | Averaging methodology | Uncertainty estimates [m] | Uncertainty source |
|---|---|---|---|
| OIB (Altimetry) | EASE-Grid 2.0 | Individual uncertainties available | Kurtz et al. (2016); Kurtz et al. (2015) |
| AEM-AWI (AEM) | EASE-Grid 2.0 | ±0.10 (SIT), ±0.10 (FRB) | SIT: Haas et al. (2007), FRB: Jutila et al. (2022) |
| ASSIST (VO) | EASE-Grid 2.0 | ±0.20 | Hutchings et al. (2018) |
| ASPeCt (VO) | EASE-Grid 2.0 | 10–20% of actual thickness | Worby et al. (2008a, 1999) |
| IMB-CRREL (IMB) | EASE-Grid 2.0 | ±0.01**** | Donald K. Perovich, Richter-Menge et al. (2006a) |
| SB-AWI (SDB) | EASE-Grid 2.0 | ±0.01 | Personal comm.,Nicolaus et al. (2021) |
| BGEP (ULS) | Temporal | ±0.05–0.10 | Krishfield and Proshutinsky (2006) |
| AWI-ULS (ULS) | Temporal | ±0.05 (summer), ±0.12 (winter)*** | Behrendt et al. (2013a) |
| TRANSDRIFT (ULS) | Temporal | ±0.05 (ULS),±0.96 (ADCP) | Belter et al. (2020), Belter et al. (2021) |
| NPI-FS (ULS) | Temporal | ±2.7E-03 (1990-1991), 1.9E-03 (1991-2003), *<br>1.3E-03 (2003-2006), 0.088E-03 (2007-2018) | Sumata (2022), supplementary materials table 3 |
| NPEO (ULS) | Temporal | ±0.05 | Morison et al. (2016)***** |
| SCICEX (ULS) | EASE-Grid 2.0 | Bias: +0.29, Unc:±0.5** | Rothrock and Wensnahan (2007); NSIDC (1998, 2006) |

VO: Visual Observation

*: Uncertainty of the monthly means Supplementary materials table 3

**: uncertainty based on 2·std

***:default: line correction for sound speed model:±0.23

****: 0.01 m accuracy in usual conditions 0.02 m if it is very cold the sensors do not work well during the melting season

*****: uncertainty estimate available from metadata





2. Measurements of total FRB are related with an overall uncertainty of $\pm 0.1$ m (Haas et al., 2007) and are given an uncertainty flag of category 2.

### 4.2.2 Stationary moorings

Although individual moorings have their own uncertainty estimates, the cause of uncertainty for observations obtained by similar measurands tend to be similar. Raw ULS measurements can be linked to significant biases caused by measuring the first return that comes from the ice closest to the sonar, which can cause draft values for deformed ice to be overestimated. They are also prone to uncertainties linked to e.g. corrections for variations in local atmospheric pressure, instrument tilts, and variations in the speed of sound in the water column (BGEP, 2003). However, all mooring data used in this validation study

has undergone some level of correction to decrease the uncertainties and biases.

BGEP, TRANSDRIFT, and NPI data have undergone significant processing resulting in no expected bias and an uncertainty in the range of $\pm 5$–10 cm for BGEP, $\pm 5$ cm for ULS data from TRANSDRIFT.

As previously mentioned NPI data was already processed into monthly means and therefore the uncertainty used in this dataset is based on the uncertainties of the monthly mean sea ice thickness product, which was created based on the monthly

mean sea ice drafts see (Sumata, 2022). In the supplementary materials of this publication (table 3) are listed four categories of uncertainties based on a mix of instrument type (ES300 or IPS4/5) and on the year. The uncertainties vary between $\pm 2.7E\text{-}03$ and $\pm 0.088E\text{-}03$. For raw NPI data the uncertainty is expected to be between $\pm 10$ cm for NPI data based on ice profiling sonars (IPS4/5 (from 2006)) and $\pm 20$ cm based on upward looking sonar (ES300) Hansen et al. (2013). Upward-looking ADCP data from TRANSDRIFT has a significantly larger uncertainty of $\pm 96$ cm, which is a consequence of ADCP's general instrument

setup being designed to measure velocity fields within the water column rather than to derive sea ice draft (Belter et al., 2021). NPEO has also undergone corrections resulting in an estimated uncertainty of $\pm 5$ cm for level and gently undulating ice, but no additional correction has been made to correct for the first return. Therefore, sea ice drafts for rough sea ice may tend to be biased high. To account for this, data points with an average sea ice draft of more than 2 meters are given an uncertainty flag of category 3, as thicker ice tends to be rougher (Tucker et al., 1992).

The mooring data for the Antarctic from the AWI-ULS dataset has undergone varying levels of processing, and the estimated uncertainty depends on both the time of the year and the applied corrections. Based on Behrendt et al. (2013a), drafts corrected by zero-line correction have an estimated uncertainty of $\pm 5$ cm in summer and $\pm 12$ cm in winter. When using the sound-speed model instead of zero-line correction, the estimated uncertainty is $\pm 23$ cm. Here we decide to use the zero-line correction result when available, as Behrendt et al. (2013a) found only a few cases where the sound-speed model performed better than

the zero-line correction. SID data from moorings 206-4 and 227-3 are given an uncertainty flag of category 3, as Behrendt et al. (2013a) states that these moorings have problems with the pressure sensor, signifying that they have undergone a simpler and likely less accurate correction. This is especially visible in the data for mooring 206-4, where some SID values in the final product have a standard deviation of more than 10 m. This data should therefore be used with care.

Behrendt et al. (2013a) also find significant biases for AWI-ULS drafts, as the measured drafts are consistently overestimated,

except when measuring on completely level ice. The bias depends on the draft depth and ice type, with MYI summer having





smaller biases of around 30 cm, whereas FYI winter has the largest biases ranging from 42 cm for ULS depth of up to 100 cm and 68 cm for ULS depth up to 180 cm. However, these biases were computed for the Arctic, and since sea ice in the Antarctic is generally younger and thinner due to e.g., differences in ocean heat flux and thermal insulation by a thicker snow cover in the Antarctic (Maksym et al., 2012; Haas, 2016), they may not be accurate. Nevertheless, by the lack of more certain estimates,
the biases presented in Behrendt et al. (2013a) (their Table 4) are used to correct the AWI-ULS SID measurements, however a dataset is also provided without the added bias correction. As the majority of Antarctic sea ice melts during summer (Haas, 2016), it is assumed here that all Antarctic sea ice is FYI, although this is not exclusively the case with especially the Weddell Sea having a larger extent of MYI (Wang et al., 2023).

### 4.2.3 Drifting buoys

IMB-CRREL drifting buoys lack information regarding the uncertainty of the data after processing. However, information about the estimated accuracy of the acoustic rangefinder sounders, used for the measurements, is provided. Therefore, this information is utilized as the uncertainty for each measurement. According to Richter-Menge et al. (2006a), the acoustic rangefinder sounders, which are located above and below the ice surface, have an accuracy of 5 mm, resulting in a combined uncertainty of 0.01 m, when summed. However, this value is likely underestimated when compared to satellite measurements,
as IMB-CRREL buoys provide localized data. Although the standard deviation of the final measurements in CCI SIT RRDP accounts for some variability, each buoy is positioned and follows its own drifting ice floe, and thus the impact of the overall variability of the ice in the area is expected to be largely unaccounted for, unless an array of buoys have been deployed which are representative of the ice on the satellite scales. Additionally, no specific uncertainty for SD versus SIT is provided, resulting in the acoustic rangefinder sounders' accuracy used as the uncertainty for both SD and SIT. Lastly, the initial SD measurement
is expected to be one of the major sources of uncertainty, but no estimate of this uncertainty is available. Due to these concerns, the uncertainty estimates of IMB-CRREL are assigned an uncertainty flag of category 3.

Uncertainty measurements are also not provided for SB-AWI. However, a study by Nicolaus and Katlein (2017) mentions that the largest source of uncertainty originates from the initial snow depth measurement, which remains unquantified. The sensor uncertainty is reported to be on a millimeter scale, with each of the four sensors linked to the snow depth buoy having
an uncertainty of 1 mm according to information from Meereisportalen (https://www.meereisportal.de/en/, last accessed on May 2, 2024). Lee et al. (2015) investigated the uncertainty of SD measurements performed with ultrasonic sensors and found that each of the three different ultrasonic sensors had an uncertainty in the range of 0.0187 to 0.0217 m. However, this study was conducted on terrestrial snow, and none of the sensors used were consistent with the one used for SB-AWI. Nevertheless, an uncertainty of 0.02 m is utilized here, as it is considered more realistic than the alternative of 1 mm. In Lee et al. (2015), a
comparison to manual snow depth measurements was also performed, revealing biases between 0.005 m and 0.1 m.

Consequently, the uncertainty estimate is based on several assumptions and does not account for time, space, or thickness variability. Therefore, SB-AWI is assigned an uncertainty flag of category 3.



### 4.2.4 Ship data

The data acquisition of ship observations from NH (collected in ASSIST) and SH (collected in ASPeCt) follow the same guidelines. Nevertheless, the uncertainty of the visual observations is not recorded as being the same. For ASSIST, the only information about uncertainty provided is the expected precision of the visual observations. The precision of estimating snow depth is not explicitly stated, but as the method for observing SIT and SD is the same, it is expected that the uncertainties will range close to the same intervals. Based on Hutchings et al. (2018), the precision of this estimate is 0.2 m for an experienced observer. ASPeCt denote that the error, when compared to drilled measurements, depends on the thickness of the ice floe (Worby et al., 2008a). For sea ice <0.1 m thick, the estimated error is ±50%; for ice between 0.1 and 0.3 m, the error is ±30%; and for level ice >0.30 m, the error is ±20%. Here, it is also stated that similar error estimates apply to snow of the same thickness. As these estimates provide a quantified uncertainty estimate, and as the data acquisition method for ASSIST and ASPeCt is the same, it is decided to use the uncertainty measures from (Worby et al., 2008a) for both. These uncertainty estimates provide some degree of variation due to the sea ice thickness dependency. Therefore, ASSIST and ASPeCt are given an uncertainty flag of category 1.

### 4.2.5 Submarine data

Bias and standard deviation of SCICEX submarine data are based on a paper by Rothrock and Wensnahan (2007) addressing the accuracy of US NAVY submarine measurements, which are a part of the SCICEX data, using all available data from 1975 to 2000. The combined estimated bias is +0.29 m when compared to the reference obtained from ice drillings, and the combined standard deviation among submarine measurements due to seven error sources is 0.25 m (see Rothrock and Wensnahan (2007) for further information). To convert this into a 95 % confidence interval an uncertainty of twice the standard deviation is used, giving a ±0.50 m uncertainty for each data point. Furthermore, the 0.29 m bias is subtracted. As the uncertainty is assumed to be the same for all data points, SCICEX data is given an uncertainty flag of category 2.

## 5 Satellite SIT CDRs

To illustrate the use of the CCI SIT RRDP reference measurements, the data has been collocated with CCI SIT CDRs v3.0 from CryoSat-2 and Envisat for both NH and SH. The satellite datasets are available from the ESA CCI open data portal (last accessed on August 8, 2024):

– CryoSat-2 (NH):https://catalogue.ceda.ac.uk/uuid/c6504378f78c4ecd9f839b0434023eff

– CryoSat-2 (SH):https://catalogue.ceda.ac.uk/uuid/861ad3c7f3a34ebd8be6f618a92bd8e3

– Envisat (NH):https://catalogue.ceda.ac.uk/uuid/92eb2ba942074bec804af6a8b5436bee

– Envisat (SH):https://catalogue.ceda.ac.uk/uuid/af96a1ec493f49caa39dc912d15f2b17





CryoSat-2 in the CCI CDR data set is available from 2010–2021 for both NH and SH, with 2010 having only data from November, December and the following years having data from October through April in the NH, and for all months in the SH. Envisat data is available from 2002–2012 for similar months as CryoSat-2.

## 5.1 Algorithm description

The method for extracting sea ice freeboard and thickness from radar altimetry data follows work of Laxon et al. (2003) and Tilling et al. (2018), where some of the key steps include distinguishing the sea ice (floes) and sea surface (leads) radar echoes, correcting for slower wave propagation speed, and calculating the sea ice thickness assuming hydrostatic equilibrium. To derive sea ice elevation estimates (and freeboards), one needs a dataset containing radar echo waveforms for range retrieval and other

relevant variables such as altitude, atmospheric and geophysical corrections, in addition to auxiliary data of mean sea surface height, sea ice type, SD, snow density and sea ice density. The CCI CryoSat-2 sea ice processing uses the Baseline D Level 1b SAR and SARIn orbit data files from November 2010 until April 2021. For Envisat, the version 3.0 of the Envisat SGDR (Sensor and Geophysical Data Record) data has been used. The auxiliary data common to both Arctic and Antarctic sea ice processing contain the DTU21 mean sea surface product (Andersen et al., 2023) and the Copernicus Climate Change Service

(C3S) CDR for sea ice concentration. For sea ice type in the Arctic the C3S CDR is used, and for the Antarctic the ice is considered to be of a single type i.e. FYI. Snow is handled for the Arctic by using the merged monthly Warren et al. (1999)-AMSR2 snow depth climatology interpolated to daily values (more in Paul et al., 2021) with the snow density modifications suggested by Mallett et al. (2020). In the SH, a revised version of the approach described by Cavalieri et al. (2014) are used. Here the daily estimated AMSR-E/2 snow depths are averaged for each calendar day of the year to form a daily climatology

used together with a fixed climatological value for snow density (Paul et al., 2021).

For CryoSat-2, the sea ice freeboard and thickness processing is done conventionally, classifying the surface type with multi-parameter approach (using the following waveform parameters: backscatter, leading edge width and pulse peakiness), and using the Threshold First Maximum Retracker Algorithm (TFMRA) with a 50% threshold from the first maximum peak power for range retrieval (find more details in Paul et al., 2021). To achieve a consistent time-series accounting for the different

types CryoSat-2 and Envisat radar altimeters, the CCI SIT CDR v3.0 Envisat product makes use of orbit crossovers and orbital overlap during coincident mission periods with CryoSat-2 during winter months between October 2010 and March 2012. This data is used to retrieve optimal retracker parameters for calibration of Envisat, while using CryoSat-2 freeboard estimates as a reference which is applied to the full Envisat period (Paul et al., 2021, 2022). The satellite data is available in two formats; a L3 gridded product and a L2 trajectory product. Here, the L2-trajectory product was used to ensure that the spatial overlap

between satellite and reference data was as close as possible. The L2 product consist of daily satellite trajectories and contain information including radar freeboard, sea ice freeboard (radar freeboard corrected for the slower radar wave propagation speed in snow), sea ice thickness and auxiliary snow depth with related uncertainties.

In addition, radar freeboards from ERS-1/2 are available within the ESA FDR4ALT project (Bocquet et al., 2023) in a gridded format for both NH and SH. As the ERS-1/2 products only include radar freeboards in its current form with no

additional information of snow depth, snow and ice densities nor sea ice types, which are needed in the radar-freeboard-to-sea-



ice freeboard and sea-ice-freeboard-to-sea-ice-thickness conversions, we only use the FDR4ALT dataset to demonstrate the availability of overlapping satellite and reference measurements during the ERS-1/2 satellite period. We note, that FDR4ALT also provides freeboards from Envisat and CryoSat-2 having, through an application of neural-networks, aimed to account for inter-satellite-mission biases caused by different acquisition modes (Bocquet et al., 2023) and thus provides a full time-series
ranging back to 1994 of radar freeboards.

## 6   Comparability and collocation of RRDP and CDR

It is imperative to ensure that we compare the same measurand of the reference measurements within the CCI SIT RRDP and the satellite altimetry derived CCI SIT CDR. From a metrological approach, the aim is to ensure that the reference measurement and the measurand of the satellite product, whether being the total FRB, sea ice FRB, SIT, SID or SD, are comparable (Da Silva
et al., 2023). Here, we aim to ensure this by, in most cases, keeping the reference measurand in its most original form and adapting the CCI SIT CDR measurand accordingly. As an example, when we compare the CCI SIT CDR with SID from ULS, we convert the satellite-derived SIT into SID by subtracting the sea ice FRB from SIT, as the ULS does not provide any information about the ice above the local sea level, following:

$$\text{SID}_{\text{CDR}} = \text{SIT}_{\text{CDR}} - \text{FRB}_{\text{CDR , sea ice}} \tag{7}$$

In addition, for NH OIB, we have coincident reference measurements of total FBR and SD, thus we compare the OIB derived sea ice FRB by subtracting the measured SD from the total FRB directly with the satellite derived sea ice freeboard, provided in the CCI SIT CDR, following:

$$\text{FRB}_{\text{OIB NH, sea ice}} = \text{FRB}_{\text{OIB NH, total}} - \text{SD}_{\text{OIB NH}} \tag{8}$$

Additionally, we compare the OIB SD directly with the auxiliary SD product in the satellite CCI SIT CDR. By using this
approach, we avoid introducing additional auxiliary products (e.g., snow depths) which is not already used, and reduce errors from introducing new products. For SH OIB, we do not have any SD measurements and the same is the case for FRB measurements from AEM-AWI. Therefore, the comparison is made by adding the auxiliary SD to the satellite derived sea ice freeboard in the CCI SIT CDR, following:

$$\text{FRB}_{\text{CDR, total}} = \text{FRB}_{\text{CDR, sea ice}} + \text{SD}_{\text{CDR}} \tag{9}$$

Similarly, the SIT measured by AEM-AWI is the total thickness (snow + ice). Hence, to compare this measurand to the satellite observations, we add the auxiliary SD product in the satellite CCI SIT CDR to the SIT in the CCI SIT CDR, following:

$$\text{SIT}_{\text{CDR, AEM-AWI}} = \text{SIT}_{\text{CDR}} + \text{SD}_{\text{CDR}} \tag{10}$$

Collocation is performed by finding all satellite data points obtained within $\pm$ 15 days from the date of the reference data, and within the 25 km (50 km for SH) grid cell of the reference coordinates. The average (arithmetic mean) of these satellite points
are subsequently allocated to the reference data.



## 7 Results and discussions

We present the comparison between CCI SIT RRDP and CDRs for each of the NH and SH sea ice variables (FRB, SIT, SID and SD) in Fig. 6–8, with the location of SH CryoSat-2 and Envisat collocated observations in Fig. 10a. The figure visualizes the geographical distribution of the CCI SIT RRDP reference measurements collocated to CCI SIT CDRs of CryoSat-2 and Envisat, respectively, together with associated scatterplots and histograms. Linear best fits are added to the scatter plots for both individual campaigns and for the combined data available for each sea ice variable. Linear fits are obtained by using an Orthogonal Distance Regression (ODR), which involves calculating the orthogonal distance of the points with respect to a linear fit and allows taking into account the errors in measurements for both independent and dependent variables (Boggs and Rogers, 1990). Hereby, the linear fits are weighted by the individual uncertainties in both the CCI SIT CDRs and the CCI SIT RRDP. The pearson correlation coefficient (R) of each fit are shown in the scatter plots and additional statistical information can be found in Tables B1 (CryoSat-2) and B2 (Envisat) in Appendix B. Histograms show the distribution of reference measurements in blue and collocated CryoSat-2 or Envisat in orange with red and black vertical lines indicating the mean of satellite and reference data, respectively. For SIT and SID a bin size of 10 cm is used, whereas 3 cm is used for SD and FRB. Data from the SH is marked by a black outline around the histogram bins. The geographical representation of both the SH sea ice variables collocated to CCI SIT CDRs of CryoSat-2 and Envisat and the NH and SH sea ice variables collocated to FDR4ALT dataset of ERS-1/2, is highly limited. Therefore, only the data locations from the variables are presented in Fig. 10, and the statistics of CryoSat-2/Envisat comparisons with reference observations are provided in Fig. 6-8. As the ERS-1/2 product only included radar freeboard at the time that this research was conducted, there has not been made any comparisons with scatter plots and histograms.

When examining Fig. 6–8, it is important to note that the main objective of this paper is not to conduct an inter-comparison study but rather to present the applications of the CCI SIT RRDP. Therefore, we focus on demonstrating how the database can be utilized to validate satellite products by highlighting the advantages and limitations of the different types of reference observations. Furthermore, we discuss the availability of reference measurements for validating the four primary variables (FRB, SIT, SID and SD) for CryoSat-2, Envisat and ERS-1/2, respectively.

## 7.1 Freeboards (FRB)

The amount of reference observations for freeboard validation are limited to airborne campaign data, which is primarily collected in March/April in NH (Sect. 2.2, Fig. 5a). Nevertheless, the airborne reference data show a good geographic representation in the western Arctic i.e., the Beaufort Sea, the Canadian Archipelago, as well as the Lincoln and Wandel Seas north of Greenland, and to a less degree the Fram Strait with CryoSat-2 and a reasonable overlap with Envisat (Fig. 6). However, no data is available from eastern Arctic including the East Siberian, Laptev, Kara and Barents Seas due to logistical challenges operating in these regions. For the SH, FRB measurements are currently limited to OIB campaign data from October 2009 and 2010 (Sect. 2.2, Fig. 5b) collected in the Weddell, Bellingshausen and Amundsen Seas (Fig. 10a). Thus, no FRB reference data is available for CryoSat-2 in SH. The FRB reference measurements, in particular those measuring the sea ice FRB, such



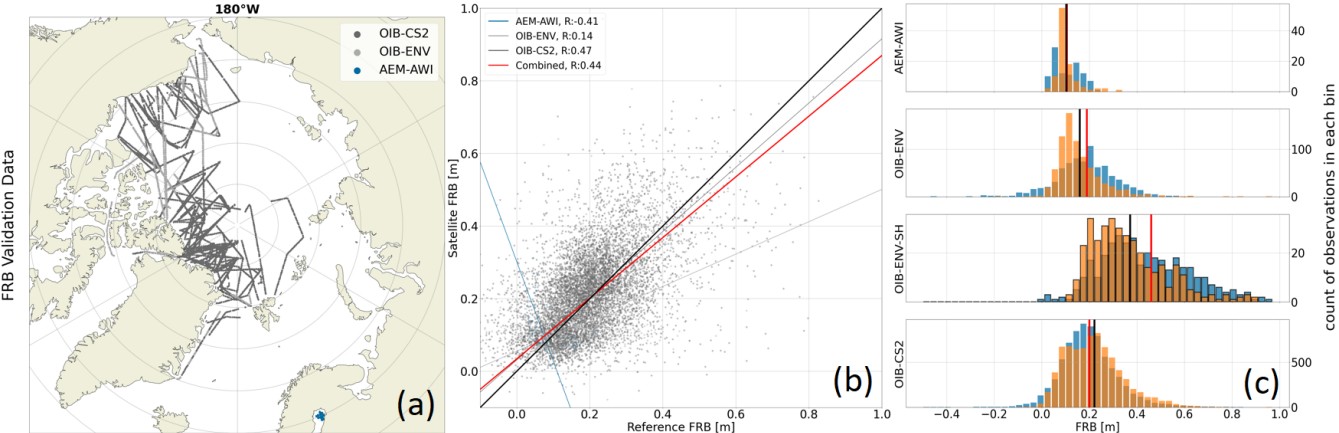

**Figure 6.** FRB satellite and reference observation comparison for CryoSat-2 (CS2) and Envisat (ENV) combined. The campaign names are followed by the satellite abbreviation, to highlight which satellite the reference data has been collocated to. (a) map of available overlap data for NH, (b) scatterplot with NH data with uncertainty weighted linear fits and associated correlation coefficients in the legend and (c) histograms with the reference observations and satellite CDRs marked with blue and orange, respectively, with associated average as red and black vertical lines. A black outline around the histogram bins are added to data from the SH.

as OIB NH, are in principle directly comparable to the sea ice FRBs from the satellite altimeter measurements. They can be used to directly evaluate the penetration depth by the radar signal and errors introduced by the reduced radar propagation speed in the snow (Section 1), assuming proper considerations regarding snow depth estimates from conversion of radar FRB to sea ice FRB, and from total FRB to sea ice FRB has been accounted for. The associated histograms in Fig. 6c show similar distributions for NH OIB and CryoSat-2 FRB with only -0.02 m mean difference (Table B1). Envisat is over-represented in thin sea ice thicknesses (FRB<0.20 m NH) and has, in general, a narrower distribution when compared to OIB FRB in NH, which is not reflected in the mean difference of 0.03 m (Table B2).

Reference data from AEM-AWI NH and OIB SH presents total FRB. Therefore, they are compared to the collocated satellite-derived sea ice FRB plus SD from the auxiliary snow depth information as provided in the CCI SIT CDR (Section 6) using Equation 9. AEM-AWI FRB only includes data from the Baltic Sea, which is reflected in the low total FRB heights (<0.3 m) representative for FYI. The limited geographical extent and the limited data amount of the AEM-AWI total FRB is expected to be one of the causes of the negative correlation coefficient ($R = -0.41$) as seen in Fig. 6b. Additionally, we do not know how well Envisat performs in the Baltic Sea, which is an area with only FYI and confined by land with many small Islands, which may impact the relatively large footprint size of Envisat. The over-representation of Envisat total FRB for thinner ice is also present in the SH when compared to OIB. The over-estimation of thin sea ice and a more narrow distribution is presumed to be a consequence of the different footprints of Envisat and CryoSat-2, and not caused by the reference measurements.



**Figure 7.** SIT satellite and reference observation comparison for CryoSat-2 (a-c) and Envisat (d-f). Maps of overlap data are shown for NH in (a,d), associated scatterplots with uncertainty weighted linear fits and associated correlation coefficients in the legend in (b,e) and histograms in (c,f). In the histograms the reference observations and satellite CDRs are marked with blue and orange, respectively, with associated average as black and red vertical lines. A black outline around the histogram bins are added to data from the SH

## 7.2 Thicknesses (SIT)

The amount of NH SIT reference measurements are more substantial than is the case for FRB and include observations from airborne campaigns, ships, and ice mass balance buoys, which complement each other in terms of spatial and temporal coverage. For CryoSat-2 (Fig. 7a) reference measurements are well covered in the western Arctic region and also include some reference measurements in the eastern Arctic north of 80°N. However, most of the reference measurements in the eastern Arc-

tic is based on visual ship observations. The reference measurements overlapping with Envisat are sparse and limited to few drifting buoys and airborne campaigns in Beaufort Sea and Fram Strait (Fig. 7d). SIT reference measurements to compare with ERS-1/2 are limited to some AEM-AWI airborne observations around Svalbard in the NH (Fig. 10b). In SH, SIT reference data is limited to visual ship observations from ASPeCt (Fig. 10a and 10c).



**Figure 8.** SD satellite and reference observation comparison for CryoSat-2 (a-c) and Envisat (d-f). Maps of overlap data are shown for NH in (a,d), associated scatterplots with uncertainty weighted linear fits and associated correlation coefficients in the legend in (b,e) and histograms in (c,f). In the histograms the reference observations and satellite CDRs are marked with blue and orange, respectively, with associated average as black and red vertical lines. A black outline around the histogram bins are added to data from the SH

A reasonable overlap is found between the distribution of IMB and CryoSat-2 SIT (Fig. 7c) with an overall bias of 0.23 m,
where IMB SIT is thicker than CryoSat-2 SIT. Nevertheless, the linear relationship between the two datasets is nonexistent, as evidenced by the low correlation coefficient (Fig. 7b) and the $R^2 \approx 0$ value (Table B1). This is mainly expected to be due to the acquisition method of the ice mass balance buoys, where each buoy is measuring the temporal evolution of the same ice floe. Hereby, the measurements are local and do not capture variations in the surrounding ice as measured on satellite scales e.g, the growth of new ice and thicker ice, and deformation caused by the divergent and convergent motion of sea ice. As the ice mass
balance buoys must be placed on a stable ice floe, this also mean that, the buoys tend to be slightly biased towards thicker ice, resulting from the need to ensure that the ice floe does not melt or deform in a manner that cause damage and premature loss of the buoy. On the other hand, the satellites measures a much larger area, which is likely why the linear relationship between the two is weak.



**Figure 9.** SID satellite and reference observation comparison for CryoSat-2 (a-c) and Envisat (d-f). Maps of overlap data are shown for NH in (a,d), associated scatterplots with uncertainty weighted linear fits and associated correlation coefficients in the legend in (b,e) and histograms in (c,f). In the histograms the reference observations and satellite CDRs are marked with blue and orange, respectively, with associated average as black and red vertical lines. A black outline around the histogram bins are added to data from the SH

The use of the visual ship observations, as reference measurements for SIT, is dubious. First, visual ship observations are dependent on human interpretation, which introduce a larger uncertainty on the individual measurements in particular if the "IceWatch" manual (Hutchings et al., 2018) is not followed in detail and thus, is subjective. In the future, more systematic methods which decreases the influence of the observer should be investigated allowing for more standardized observations. Additionally, as stated in U.S.Fleet (2007, Chapt. 7), icebreakers tend to choose the fastest and most economical route, which usually means avoiding ice to the highest degree possible by navigating in areas with thin ice, in leads or where there is low concentration. Additionally, Hutchings et al. (2018) states that only level ice should be recorded due to the likelihood of thicker ice not fully overturning. The two factors combined suggests that observations from ships tend to have a larger representation of thin ice in the SIT distribution.





This tendency is clearly reflected in the ASSIST and ASPeCt SIT distributions when compared both to CryoSat-2 and Envisat (Figs. 7c and 7f). This results in large negative biases of -0.47 m and -0.77 m, when comparing ASSIST and ASPeCt
SIT to CryoSat-2 SIT. Similar large negative bias of -0.63 m is found between ASPeCt and Envisat SIT, whereas the bias is smaller for ASSIST (-0.17 m). This is expected to be partly due to the limited amount of available ASSIST reference data overlapping with Envisat (Fig. 7d). As shown in Table B2 96 data points are available in the CCI SIT RRDP after averaging. In general, few ship observations are averaged for each value in the CCI SIT RRDP, as a result of limited data availability for each 25 km (50 km) grid cell for NH (SH). As a reference, 1.1 observations are used per average for the ASSIST measurements
coinciding with Envisat and 1.95 for CryoSat-2. This means that the 96 available reference data points from ASSIST to Envisat represent a limited amount of information, and is thus related with a larger uncertainty in the reference measurements.

### 7.3 Sea ice draft (SID)

SID data has a very limited geographical representation, as seen in Fig. 9a for CryoSat-2 and Fig. 9d for Envisat, and Fig. 10b for ERS-1/2 in NH, as most of SID data (except for SCICEX) is obtained from stationary moorings. However, the moorings
represent a time-averaged SID and thus represent a larger sample of sea ice due to the sea ice drift (Sect. 2.3), and have been used extensively to validate satellite-derived sea ice thicknesses (e.g. Sallila et al., 2019; Quartly et al., 2019). A good agreement are seen in the distributions (Fig. 9c) between CryoSat-2 and SID data from SCICEX, NPI and BGEP.

The TRANSDRIFT data is the only sea ice reference measurement available for the Seas north of Russia, and represents data over assumed fairly level FYI (the ideal sensing scenario for CryoSat-2). Nevertheless, the difference in the distributions
between CryoSat-2 and TRANSDRIFT SID is larger then those of SCICEX, NPI and BGEP. In Belter et al. (2020) they find a correlation coefficient ($R = 0.47$) and a mean difference (0.28 m) using orbit information from CryoSat-2 similar to our approach. These values are similar to those found in this study ($R = 0.52$) and mean difference of (0.33 m). Their findings of a general tendency of the satellite CDRs to overestimate SID <0.7 m and underestimate SID >1.3 m are in agreement with our findings. The SID distributions for Envisat and reference measurements in NH (Fig. 9f) are similar to those discussed for
Envisat in SIT Sect. 7.2, where Envisat tend to be over-represented in the thinner ice and vise versa for thicker ice. This is most pronounced for the comparison with SCICEX and TRANSDRIFT SID, where the reference observations have SID up to 3 m and Envisat SID < 1.8 m. SID data from NPEO is also available in the CCI SIT RRDP, but this data is located at the north pole and is, therefore, within the polar gap of the existing satellites. Nonetheless, the data has been included to validate future satellite products which may cover the polar gap by the implementation of different interpolation techniques. The strength of
the SID reference measurement record is that NH NPI-ULS and SCICEX covers both CryoSat-2, Envisat and ERS-1/2.

The representation of SID data for SH (Fig. 10a and 10c) is limited to moorings in the Weddell Sea and Lazarev Sea, which only overlaps with Envisat and ERS-1/2. This data shows a reasonable overlap in the distributions with AWI-ULS SID (Fig. 9f) being on average 0.21 m higher when compared to Envisat SID if no bias correction is applied, whereas application of a bias correction results in an average bias of Envisat SID being 0.06 m higher than AWI-ULS SID. However, no significant trend
($R^2 \sim 0$) is seen between Envisat SID and AWI-ULS SID (Table B1).





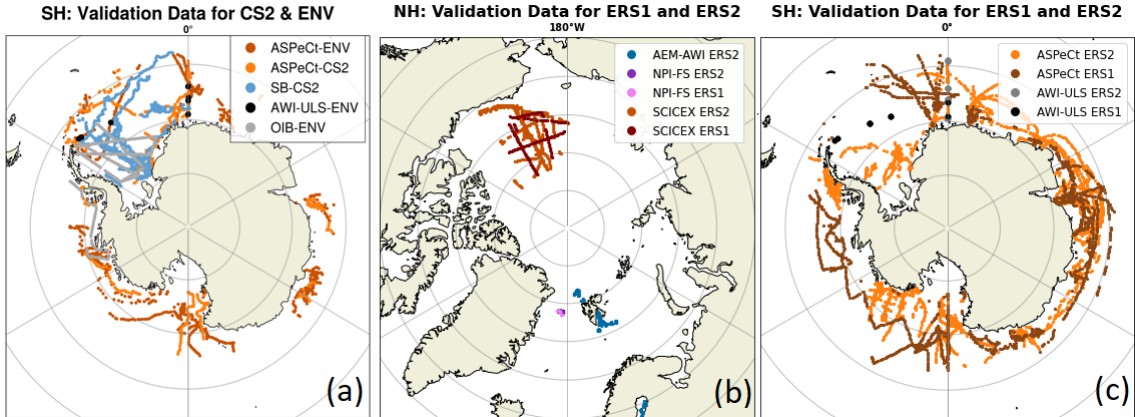

**Figure 10.** Maps showing the geographical representation of; (a) SH reference measurements collocated with CryoSat-2 and Envisat data, (b) NH reference measurements collocated with ERS-1/2 and (c) SH reference measurements collocated with ERS-1/2.

## 7.4 Snow depth (SD)

A quite extensive amount of SD reference observations are available for the NH in particular overlapping with CryoSat-2 (Fig. 8a), and to a lesser extent Envisat (Fig. 8d), including airborne, buoy and ship data. The geographic distribution of data is similar to the SIT data presented in Sect. 7.2. For SH, ASPeCt ship observations are available as presented in Fig. 10a for CryoSat-2 and Envisat, and in Fig. 10c for ERS-1/2. Snow depth measurements from buoys (SB-AWI-SH) are only available for CryoSat-2.

The comparison of the distributions presented in Figs. 8c and 8f show large variability, where the CCI SIT CDRs auxiliary snow depth climatologies tend to have a narrower distribution when compared to the different reference observations. This indicates that the auxiliary snow depth climatologies provided in the CCI SIT CDRs (described in Sect. 5.1) fail to quantify sufficient variability in the snow depth distribution, when compared to reference data, even for the SH snow depth climatology, which is based on daily satellite radiometer observations. This is a fair assumption, as climatologies are averages and do not provide inter-annual variations. However, the variability may also be caused by the reference data having too high variability, which may not signify the larger scale variability of satellite products. This was also discussed in Stroeve et al. (2020a) for drifting buoy observations of snow depths compared to models of similar spatial resolution (25 km) as for the CCI SIT CDRs in NH. Further work should be prioritized to improve this.

Nevertheless, reasonably high correlation ($R > 0.45$, Figs. 8b and 8e) coefficients are obtained for several reference data sources such as, ASSIST and OIB. Therefore, although the distributions do not show strong agreement, some degree of linear correlation is observed between the satellite SD product and certain, but not all, sources of reference observations.





## 7.5 Uncertainty quantification

Accurate uncertainty quantification is crucial to obtain the full potential of reference observations. However, this requires individual uncertainty measurements, that are ideally propagated all the way from the raw measurements through the processing chain to the final estimates. In this study uncertainties have been assumed to be independent, although this is likely not the case, and is expected to result in an underestimation of the uncertainties for the CCI SIT RRDP. However, in lack of more accurate estimates these values provide a first assessment of uncertainties and can be used as an alternative to the uncertainty values provided in Table 4. In the future, all reference measurements (buoy, airborne, submarine and ship) should be quantified with individual uncertainty measurements or provided with uncertainty estimates that take into account some degree of seasonal/spatial variation.

An additional uncertainty source in the CCI SIT RRDP arises from the representativeness of the reference data for a 25/50 km grid cell for a monthly period. An attempt has been made to provide an estimate of this by providing the standard deviations and number of entries for each grid cell. However, differences in spatial and temporal coverage between reference data sources and between satellite products within a 25/50 km grid cell for a monthly period might result in temporal and spatial biases, which are not accounted for here. However, this study provides an initial assessment and steps towards a unified uncertainty estimate for SIT reference observations, which requires future iterations and additional information to be fully described.

## 7.6 Filtering, representativeness and stability

Input data to the CCI SIT RRDP has not been filtered and therefore data shown in this paper include observations with high standard deviations and/or observations based on very few input measurements. To maximize the usability of the CCI SIT RRDP the user should consider removing observations with very high standard deviations or very few input measurements. If the standard deviation is very high, the data either contains outliers that could be faulty or the variation of the reference data within the grid cell is very high, reflecting a complex ice topography within the grid cell or conditions where the sea ice have undergone large variations within the time-span of the reference observation (up to a month) within the grid cell e.g., caused by deformation. On the other hand, very few reference measurements included in the grid cell estimate could indicate that the reference data will not be representative of the whole area and/or time period. Nevertheless, these considerations are highly linked to measurement and method as the number of e.g., airborne measurements are typically high although the data is typically limited to a few days in a specific region, which is compared to a monthly CDR mean value, questioning the representativeness. However, when comparing CryoSat-2 NH observations with OIB sea ice FRB ($R = 0.47$, Fig. 6b), AEM-AWI total SIT ($R = 0.59$, Fig. 7b), OIB derived SIT ($R = 0.56$, Fig. 7b) and OIB SD ($R = 0.57$, Fig. 8b) we find relatively high correlation coefficients, which are similar to other comparison studies using airborne campaign data for monthly satellite evalutions (e.g. Quartly et al., 2019; Sallila et al., 2019). Other methods, such as matching the satellite SIT CDRs and reference measurements from drifting buoys and stations in a Lagrangian framework i.e., following the buoys, might be a more suitable methodology for future comparisons. This approach was successfully demonstrated by (Stroeve et al., 2020a) where they





compare snow depths from the CRREL-IMBs and snow depth buoys to a snow depth model at similar spatial scales to NH satellite products in our study.

To test the stability of the satellite-based SIT CDRs with reference observations we here recommend using a combination of the long-term monitoring programs i.e., upward-looking moorings from the Beaufort Gyre Exploitation Project (BGEP), the Fram Strait Arctic Outflow Observatory (NPI-FS) and the Russian-German TRANSDRIFT project (TRANSDRIFT), together with submarine cruises (SCICEX) and airborne EM-campaigns (AEM-AWI) for NH. A similar extensive dataset is not found available for the SH in particular with the biased ASPeCt sea ice thickness observations. To combine the NH measurements into a trend dataset, we need to align the reference measurements and the satellite CDRs to a common measurand e.g., total

SIT or draft. For consistency we shall ensure that a common baseline for auxiliary information of snow depth and densities of sea ice, snow and water are used. Thus, we encourage the community to include the information of all auxiliary data used to support the radar-freeboard-to-sea-ice-freeboard and sea-ice-freeboard-to-thickness conversions to form a consistent baseline.

## 8   Code and data availability

The CCI SIT RRDP dataset is available at DTU DATA (https://figshare.com/s/77be0cfd6842d08f1b6b) (Olsen and Skourup,

2024a) and released under a CC-BY 4.0 license. The final data files are formatted as NetCDF files with not-a-number (NaN) denoting missing data. Apart from the final dataset, all the source code used in the processing steps from the original reference observations to the final CCI SIT RRDP, together with procedures for collocating the CCI SIT RRDP to the satellite measurements from CryoSat-2, Envisat and ERS-1/2 are available on GitHub through (Olsen and Skourup, 2024b). Figure 2 (Sankey diagram) was produced using the online tool "VisualParadigmOnline", and adapted in Microsoft Office Power Point. Links to

the original reference observations and the satellite data are provided in Table 2.

## 9   Conclusions

Here, we have presented the CCI SIT RRDP (Olsen and Skourup, 2024a) of available sea ice thickness reference observations covering the polar satellite era 1993–2021 including freeboards (total, radar or derived sea ice), thicknesses (total or sea ice), drafts and snow depths from different sources. The CCI SIT RRDP is suitable for the evaluation of satellite altimeter

observations of sea ice freeboard, sea ice thickness and auxiliary snow depth products, but can also be used for evaluation of e.g. models. The reference observations have been prepared to a level where they can be directly compared to satellite altimetry temporal (monthly) and spatial (25 km NH; 50 km SH) scales, but these can easily be changed by using the CCI SIT RRDP software package Olsen and Skourup (2024b). We have added uncertainties to the associated reference measurements and flagged these according to their reliability.

As examples of how this data package can be used, we have compared them with the CCI SIT CDRs from CryoSat-2 and Envisat. Here, we generally find good agreement across the different reference observations. Visual observations of sea ice thickness and snow depths from ship cruises cannot be used as reference measurements for satellite altimetry in its present





form presented in the RRDP, as they are biased low in their distributions. This is expected as ships tend to navigate through the thinnest ice. In addition, the approach used in this study by gridding and time-averaging the reference measurements from
drifting buoys and stations to match satellite scales might not be the most optimal. Other methods, such as matching the satellite SIT CDRs and reference measurements from drifting buoys and stations in a Lagrangian framework might present a more suitable solution in the future. However, existing reference data is scarce in the polar regions, even more pronounced in the Antarctic than in the Arctic. It is therefore necessary to include and use as much data as possible while acknowledging the advantages and limitations of each method. We also note that data prior to 2011 is limited to a few reference measurements
- even more so if we remove observations above the polar gap - thus, limiting the comparisons to be made with the ERS-1/2 satellites. Existing reference measurements, which are not included in the CCI SIT RRDP, need to be made publicly available and processed to a level where they can directly be used for inter-comparison to satellite altimetry-derived observations before inclusion into the CCI SIT RRDP. This includes e.g., freeboards from ESA's CryoSat Validation Experiment (CryoVEx) campaigns, as well as OIB Antarctic total freeboards and snow depths. When satellite altimetry-derived sea ice thicknesses are
provided in a CDR it is important for comparison and evaluation purposes to include all the auxiliary information used in the intermediate steps in the radar freeboard to sea ice thickness conversion.

For future work, it will be crucial to ensure that reference observations follow the protocols and procedures for fiducial reference measurements (FRMs) i.e., that they are traceable, and fully described with uncertainty diagrams, effects tables and comparability diagrams, see first efforts for altimetry derived sea ice thicknesses in Da Silva et al. (2023). Such comparabil-
ity diagrams will also aid the design of campaigns to produce measurements which are directly comparable to satellite SIT products. As an example, the AWI IceBird Winter campaigns since 2019 use a sensor combination of EM-Bird, airborne laser scanner and snow radar that allows for a direct retrieval of sea ice thickness, sea ice freeboard and snow depth simultaneously (Jutila et al., 2022). Snow depths from these campaigns have recently been made available (Jutila et al., 2021) and should be included in future iterations of the CCI SIT RRDP. Additionally, recently published data sources, such as the Nansen Legacy
project's snow depth and sea ice thickness dataset in the Barents Sea and Arctic Basin (Divine, 2023a, b), as well as those from the MOSAiC Expedition, will be included in future iterations of the CCI SIT RRDP. These datasets will provide data in areas that are underrepresented in the current version. Updating the data package will be ongoing work to ensure it remains current and comprehensive. The community also urgently need to ensure a consistent network of polar observations for continuous reference measurements of current and future satellite altimeter missions such as Copernicus Polar Ice and Snow Topography
Altimeter (CRISTAL) (Kern et al., 2020).

*Author contributions.* IO has collected and prepared the CCI SIT RRDP presented in this paper including pre-processing, estimation of uncertainties and uncertainty flags. The CCI SIT RRDP is based on previous versions prepared by HSK within the initial phases of the CCI SI project. The initial versions included only sea ice freeboards and snow depth with no uncertainties nor uncertainty flags provided until 2016, https://ftp.spacecenter.dk/pub/SICCI/. IO further collocated the reference data with the satellite CDRs. IO, HSK, HS, RFH wrote
the manuscript. HSK, ER, SH, SK contributed by identifying relevant reference measurements, defined the temporal and spatial resolution





together with the collocation procedures. They also contributed through discussions. HSK, ER, SK contributed to the development of the first phases of the CCI SIT RRDP including validation. SH, SP, ER and HS contributed to satellite derived CCI SIT CDRs from CryoSat-2 and Envisat. MB and SF contributed with ERS-1/2 data from the FDR4ALT project. DD contributed with NPI mooring data.

*Competing interests.* The authors declare that they have no conflict of interest.

*Acknowledgements.* This publication was funded by the ESA's Climate Change Initiative (CCI) for sea ice (grant no. 4000126449/19/I-NB). The main contribution to this work was conducted while I. B. L. Olsen was affiliated with DTU Space. The project and its results are primarily associated with this institution. I. B. L. Olsen is now affiliated with DMI, which contributed to the revision process of the manuscript. We would like to thank DMI for their support during this phase.

We would also like to express our sincere gratitude to everyone involved in the collection, preparation, maintenance, and publication
of the input data used in this study. These include, but are not limited to, the Beaufort Gyre Exploration Program based at the Woods Hole Oceanographic Institution (https://www2.whoi.edu/site/beaufortgyre/) in collaboration with researchers from Fisheries and Oceans Canada at the Institute of Ocean Sciences, the Norwegian Polar Institute, the North Pole Environmental Observatory and the Polar Science Center, the Russian-German BMBF-funded TRANSDRIFT project, the Alfred Wegener Institute, the Norwegian Meteorological Institute, the Cold Regions Research and Engineering Laboratory, the SCAR Antarctic Sea Ice Processes and Climate (ASPeCt) program
(aspect.antarctica.gov.au), the U.S. Navy and Royal Submarines, and the Submarine Arctic Science Program. We acknowledge that it would not have been possible to create the Climate Change Initiative Sea Ice Thickness Round Robin Data Package without the up-to-date and publicly available reference data from the above sources.

We further acknowledge the important contributions of the CCI Sea Ice scientific leader, Thomas Lavergne, and project coordinator, Mari Anne Killie, both from the Norwegian Meteorological Institute (METNO). Their exceptional leadership, management, and insightful
discussions have been pivotal to the success of this study.





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



# Appendix A:  Acronym table

**Table A1.** Acronym table - for Acronyms given to campaigns included in the CCI SIT RRDP, see Table 1.

| Abbreviation | Definition |
| --- | --- |
| ATM | Airborne Topographic Mapper |
| CCI | Climate Change Initiative |
| CDR | Climate Data Record |
| CRISTAL | Polar Ice and Snow Topography Altimeter |
| CryoVEx | CryoSat Validation Experiment |
| EASE2 | Equal-Area Scalable Earth grid, version 2 |
| ECV | Essential Climate Variable |
| EM | Electromagnetic |
| Envisat | Environmental Satellite |
| ERS-1/2 | European Remote Sensing Satellites 1 and 2 |
| ESA | European Space Agency |
| FDR4ALT | Fundamental Data Records for Altimetry |
| FRB | Freeboard |
| FRM | Fiducial Reference Measurements |
| GNSS | Global Navigation Satellite System |
| ICESat | Ice, Cloud and land Elevation Satellite |
| LiDAR | Light Detection and Ranging |
| NASA | National Aeronautics and Space Administration |
| NH | Northern Hemisphere |
| NSIDC | National Snow and Ice Data Center |
| RA | Radar Altimeter |
| RRDP | Round Robin Data Package |
| SAR | Synthetic Aperture Radar |
| SH | Southern Hemisphere |
| SD | Snow Depth |
| SID | Sea Ice Draft |
| SIRAL | SAR interferometric radar altimeter |
| SIT | Sea Ice Thickness |
| ULS | Upward Looking Sonar |
| WGS | World Geodetic System |



**Appendix B: Statistics of comparison between satellite and reference data**

The following tables show statistics linked to the comparison of CCI SIT RRDP to Envisat CDR and CryoSat-2 CDR. The comparison includes the standard deviation (std) and average of reference and satellite data, respectively. The Root Mean Square Error (RMSE), Pearson correlation coefficient (R) ( only shown in plots 6–8) and Coefficient of determination ($R^2$) for both the best fit and for the ($y = x$) fit, as would be the ideal case.

Due to the nature of R2 being a calculation between a true value and a fit this calculation is highly sensitive to which variable is being examined. In the case below the ($R^2$) value is calculated between the satellite CDRs and the predictions.





**Table B1.** Statistics of results CryoSat-2, LS= Least Squares, ODR= Orthogonal Distance Regression

| Campaign | variable | Avg. (obs) [m] | Avg. (CS-2) [m] | std. (obs) [m] | std. (CS-2) [m] | bias (obs-sat) [m] | $R^2$ ODR | RMSE ODR [m] | $R^2$ LS | RMSE LS [m] | $R^2(y=x)$ | RMSE$(y=x)$ | points | ODR fit |
|---|---|---|---|---|---|---|---|---|---|---|---|---|---|---|
| BGEP | SID | 1.17 | 1.13 | 0.48 | 0.43 | 0.03 | 0.46 | 0.32 | 0.47 | 0.31 | 0.28 | 0.37 | 160 | y=0.72x + 0.32 |
| NPI-FS | SID | 1.59 | 1.69 | 0.42 | 0.87 | -0.10 | -7243.27 | 73.85 | 0.01 | 0.86 | -0.34 | 1.00 | 155 | y=175.13 x + -276.48 |
| SCICEX | SID | 2.20 | 2.30 | 0.57 | 0.60 | -0.10 | -0.23 | 0.71 | 0.03 | 0.59 | -0.60 | 0.76 | 316 | y=0.71x + 0.51 |
| TRANSDRIFT | SID | 1.12 | 0.79 | 0.49 | 0.23 | 0.33 | 0.25 | 0.20 | 0.27 | 0.20 | -4.19 | 0.53 | 76 | y=0.32 x + 0.43 |
| AEM-AWI | SIT | 3.25 | 3.12 | 1.16 | 1.38 | 0.13 | -0.12 | 1.59 | 0.29 | 1.17 | 0.18 | 1.25 | 355 | y=1.39x -1.05 |
| ASSIST | SIT | 0.50 | 0.97 | 0.43 | 0.72 | -0.47 | -0.02 | 0.73 | 0.13 | 0.67 | -0.36 | 0.84 | 679 | y= 1.24 x + 0.31 |
| IMB-CRREL | SIT | 1.71 | 1.48 | 0.75 | 0.68 | 0.23 | 0.01 | 0.70 | 0.02 | 0.67 | -1.05 | 0.97 | 1400 | y= 0.18 x + 0.99 |
| OIB | SIT | 2.79 | 2.42 | 1.25 | 0.89 | 0.37 | 0.02 | 0.89 | 0.31 | 0.74 | -0.56 | 1.12 | 7634 | y= 0.79 x + 0.28 |
| SB-AWI | SD | 0.43 | 0.27 | 0.22 | 0.08 | 0.17 | -0.8 | 0.29 | 0.01 | 0.08 | -11.05 | 0.29 | 761 | y= -0.44 + 0.43 |
| ASSIST | SD | 0.10 | 0.13 | 0.10 | 0.08 | -0.04 | -23.71 | 0.52 | 0.35 | 0.06 | -0.29 | 0.09 | 481 | y= 4.38 x + 0.05 |
| IMB-CRREL | SD | 0.31 | 0.23 | 0.25 | 0.08 | 0.08 | -0.44 | 0.09 | 0.00 | 0.08 | -11.56 | 0.27 | 1866 | y= 0.21 x + 0.16 |
| OIB | SD | 0.24 | 0.30 | 0.10 | 0.08 | -0.07 | -1.07 | 0.12 | 0.33 | 0.06 | -0.91 | 0.11 | 7668 | y= 1.37 x + 0.04 |
| OIB | FRB | 0.20 | 0.22 | 0.13 | 0.12 | -0.02 | 0.02 | 0.12 | 0.22 | 0.11 | -0.14 | 0.13 | 7668 | y= 0.88 x + 0.03 |
| ASPeCt | SIT | 0.61 | 1.38 | 0.45 | 0.65 | -0.77 | -0.06 | 0.67 | 0.01 | 0.65 | -1.68 | 1.07 | 523 | y= 0.56 x + 1.08 |
| ASPeCt | SD | 0.19 | 0.19 | 0.19 | 0.07 | 0.00 | -26.72 | 0.46 | 0.10 | 0.07 | -5.57 | 0.18 | 523 | y= 2.03 x + 0.08 |
| SB-AWI-SH | SD | 0.73 | 0.25 | 0.86 | 0.08 | 0.48 | -0.24 | 0.11 | 0.04 | 0.08 | -152.02 | 1.00 | 821 | y=-0.07x + 0.23 |





**Table B2.** Statistics of results Envisat, LS= Least Squares, ODR= Orthogonal Distance Regression

| Campaign | variable | Avg. (obs) [m] | Avg. (ENV) [m] | std. (obs) [m] | std. (ENV) [m] | bias (obs-sat) [m] | $R^2$ ODR | RMSE ODR [m] | $R^2$ LS | RMSE LS [m] | $R^2(y=x)$ | RMSE$(y=x)$ | points | ODR fit |
|---|---|---|---|---|---|---|---|---|---|---|---|---|---|---|
| BGEP | SID | 1.46 | 1.05 | 0.59 | 0.41 | 0.41 | 0.46 | 0.30 | 0.56 | 0.27 | -0.92 | 0.57 | 184 | y= 0.71x + 0.07 |
| NPI-FS | SID | 2.16 | 1.66 | 0.65 | 0.71 | 0.49 | -1.25 | 1.07 | 0.05 | 0.69 | -0.91 | 0.98 | 141 | y=1.49x -1.56 |
| SCICEX | SID | 1.61 | 1.07 | 0.77 | 0.41 | 0.54 | 0.43 | 0.31 | 0.51 | 0.28 | -2.66 | 0.77 | 228 | y= 0.53x + 0.22 |
| TRANSDRIFT | SID | 1.43 | 0.85 | 0.70 | 0.31 | 0.58 | 0.50 | 0.22 | 0.53 | 0.22 | -5.10 | 0.78 | 55 | y= 0.40 x + 0.30 |
| AEM-AWI | SIT | 1.94 | 1.51 | 1.09 | 0.89 | 0.43 | -0.19 | 0.97 | 0.14 | 0.82 | -0.83 | 1.20 | 548 | y= 0.77 x + 0.08 |
| ASSIST | SIT | 0.42 | 0.59 | 0.46 | 0.19 | -0.17 | -1.73 | 0.38 | 0.01 | 0.19 | -6.25 | 0.51 | 98 | y= 0.58 x + 0.13 |
| IMB-CRREL | SIT | 1.89 | 1.25 | 0.90 | 0.71 | 0.64 | 0.02 | 0.73 | 0.02 | 0.70 | -2.02 | 1.24 | 1086 | y= 0.17 x + 0.75 |
| OIB | SIT | 2.63 | 1.88 | 1.45 | 0.82 | 0.74 | -1.18 | 1.22 | 0.04 | 0.80 | -3.24 | 1.68 | 780 | y= 0.74 x + 0.11 |
| ASSIST | SD | 0.05 | 0.12 | 0.04 | 0.04 | -0.07 | -9.86 | 0.18 | 0.21 | 0.04 | -2.99 | 0.08 | 96 | y= 4.22 x + 0.04 |
| IMB-CRREL | SD | 0.30 | 0.18 | 0.21 | 0.07 | 0.13 | -0.16 | 0.08 | 0.08 | 0.07 | -9.07 | 0.24 | 1401 | y= 0.28 x + 0.10 |
| OIB | SD | 0.19 | 0.25 | 0.09 | 0.09 | -0.06 | -0.88 | 0.13 | 0.34 | 0.07 | -0.42 | 0.10 | 795 | y= 1.54 x + 0.01 |
| AEM-AWI | FRB | 0.11 | -0.00 | 0.06 | 0.04 | 0.11 | -7.63 | 0.17 | 0.17 | 0.08 | -11.16 | 0.13 | 101 | y= -2.74x + 0.30 |
| OIB | FRB | 0.19 | 0.16 | 0.16 | 0.11 | 0.03 | -0.23 | 0.13 | 0.02 | 0.11 | -1.73 | 0.18 | 795 | y= 0.45 x + 0.06 |
| AWI-ULS | SID | 1.27 | 1.33 | 0.82 | 0.91 | -0.06 | -3.18 | 2.21 | 0.06 | 0.89 | -0.38 | 1.07 | 214 | y= 2.27x - 0.35 |
| ASPeCt | SIT | 0.72 | 1.36 | 0.51 | 0.75 | -0.64 | -0.01 | 0.75 | 0.03 | 0.74 | -0.99 | 1.05 | 747 | y= 0.50 x + 0.93 |
| ASPeCt | SD | 0.21 | 0.19 | 0.18 | 0.09 | 0.02 | -0.95 | 0.13 | 0.20 | 0.08 | -2.58 | 0.16 | 750 | y= 0.72x + 0.08 |
| OIB-SH | FRB | 0.46 | 0.37 | 0.22 | 0.18 | 0.09 | 0.19 | 0.16 | 0.29 | 0.15 | -0.41 | 0.21 | 364 | y= 0.70x + 0.05 |