# Peer review of "Dual-hemisphere sea ice thickness reference measurements from multiple data sources for evaluation and product inter-comparison of satellite altimetry"

_Earth System Science Data, 2024_

## Author Comment (AC1)

**Response to referee**

ilo and hsk

January 2025

We thank the reviewer, Alek Petty, for the time, effort, and expertise spent reviewing our paper. Please find below reviewer comments provided in black, and our comments in red.

**1 Answer to review by Alek Petty - General review**

The paper by Olsen et al., introduces a compiled dataset of sea ice thickness related 'reference' measurements (freeboard, ice draft, snow depth, sea ice thickness) from various sources towards the goal of validating satellite-derived (radar) products across both poles through an ESA Climate Change Initiative project. They aim to align the various data with monthly gridded (25/50 km) satellite grid-scales to more easily enable evaluations. The authors make the claim in the abstract (and similar statements in the main manuscript) that this is "the first published comprehensive collection of sea ice reference observations including freeboard, thickness, draft and snow depth from sea ice-covered regions in the Northern Hemisphere (NH) and the Southern Hemisphere (SH)".

Overall, I think this was a decent effort to compile various sea ice datasets of interest, but I was ultimately disappointed with how basic the methodology was for processing the different datasets and accounting for the different uncertainties and significant differences in spatial scales (representation errors) that I remain unsure how useful this 'reference' catalog will really be. It also didn't include a lot of the more recently available data I was expecting to see. Our community hasn't produced an agreed upon 'reference' data collection as it's very hard to do this and be consistent with the uncertainties and include a full accounting of things like representation/sampling error, and it often depends on the exact goal of the validation effort.

If your primary goal is to bring in datasets that measure sea ice at vastly different spatial/temporal scales to convert these into 'reference' measurements to validate (gridded) satellite products, then you really need to consider how best to do that. I know a lot of studies just bin data into a grid-cell (myself included), but if this paper is focused on creating a reliable/useable reference processed dataset, then I think you need to acknowledge when this works and

when it doesn't and ideally explore better ways of doing that through more sophisticated statistical means.

The reference dataset produced in this study was designed for quality control of the ESA Climate Change Initiative (CCI) sea ice thickness (SIT) climate data record (CDR), prioritizing simplicity due to the many unknowns in the field. Rather than assuming that more sophisticated statistical methods would resolve these uncertainties, we believe that a simple approach is desirable given the complexity of the uncertainty budget, including sampling bias and thickness conversions. The discussion of how to treat the uncertainty budget is peaking, and it is a highly complex topic without an agreed-upon consensus. As the reviewer himself mentions, the community has not yet established a consensus on a reference dataset or standardized guidelines for the processing of reference observations, further reinforcing the need for a straightforward methodology at this stage. We are in a new era where the community is focusing more on defining the framework of Fiducial Reference Measurements (FRMs). The authors are involved in several ESA projects, that are in the early stages of developing a more thorough and sophisticated approach to define and handle FRMs in terms of best practices, including uncertainty diagrams, time-space averaging, representativeness, and error propagation. This is ongoing work and we are yet to know the outcome of these projects. Given the uncertainties, such as the representation error, which you highlight, we believe that it is advisable to use a simple approach (which the reviewer also notes that he utilises) until more ideal methods are possibly discovered in these projects. However, there is yet no proof that "more sophisticated statistical means" will address the underlying issues of the complex uncertainty budget of sampling bias and thickness conversions.

To our knowledge, there is currently only one published paper that discusses initial thoughts on how to treat FRMs for satellite altimetry over sea ice i.e., Da Silva et al. 2023, which we reference in the discussion section. Additionally, some dedicated data products, e.g., AWI IceBird, are now beginning to incorporate and provide uncertainty estimates in their newer products (available in data from 2017 and 2019 campaigns used in Jutila et al. 2022), even though they, currently, still rely on the constant single values for AEM, which are those we refer to in Table 4 in our paper.

Thus, we believe that waiting to publish until all uncertainties have been resolved, all error propagations and averaging protocols have been identified, and best practices have been defined or resolved, would not be in the community's best interest. Our method represents a first approach - but not the only approach - currently relatively widely used (as suggested by the reviewer himself) by the broader community. To allude to this aspect, we propose to change the title to "A first approach to dual-hemisphere reference measurements from multiple data sources for evaluation and product inter-comparison of satellite altimetry over sea ice".

Furthermore, by making this dataset publicly available, we contribute not only with the data processed to a format widely used by the community, but also provide a processing pipeline that can use the originally published reference observations in their native format and performs all the pre-processing steps as

described in Section 3 (this step needs to be done by all users, and thus every user shall track down the documentation and apply the necessary steps), add uncertainty estimates, flags, and finally time-space average to a level which is comparable to satellite scales. By providing the code for the processing pipeline (given in the "Code availability section" available on GitHub (Ida Olsen and Henriette Skourup 2025)), the user can easily accommodate and integrate alternative time-space averaging or apply new methods for uncertainty estimates. To our knowledge, this is the first time such a dataset has been published with added uncertainty estimates and a preliminary method for handling flags.

It is worth mentioning that the Sea Ice-thickness product iNter-comparison eXerciSe (SIN'XS) project (`https://sinxs-tools.noveltis.fr/`), which is much larger in scope than ours, also incorporates reference measurements and data comparisons using monthly gridded data – showcasing, that this methodology is currently the go-to in the community, and while newer and optimal solutions could be sought, we are not aware of many studies that have conducted such sensitivity studies and proposed new approaches, definitively. We also do not claim our approach to be conclusive; we recognize its limitations, as you have pointed out. To address the concerns raised by the reviewer, we propose to create a second version of the dataset employing a more restrictive approach to data inclusion. For example, we suggest re-evaluating the flags to be able to exclude data that we do not recommend to be used e.g., the ship observations, where the number of measurements is below a certain threshold, or where the standard deviation remains below a pre-defined threshold (whenever reasonable and applicable). We would further provide the statistics by including all the observations, and compare to statistics by excluding those with flags of different categories. In this way, we provide the statistics to support our recommendations for which classes based on the flags to include and based on this the user can easily use these flags to identify the most optimal data for their purpose while maintaining all the data in the RRDP.

As for exploring more "sophisticated" methods, this is unfortunately beyond the scope of this study, although we are keen to know which particular methods you have in mind and would encourage you to provide references for us to include in the discussion of the paper. It would be a natural extension of this work to investigate different averaging approaches, such as e.g., the Lagrangian approach proposed in Section 7.6.

We suggest extending the already included reference observation time-series upon availability. Additionally, we propose including available reference observations of similar types to those already included, such as HEM from MOSAiC, Nansen, and N-ICE. However, new types of reference observations such as drone measurements, will not be included, as we aim to maintain long time-series of consistent measurement types for quality control of the satellite CDR. We would greatly appreciate if you could share any other datasets you have in mind.

In a lot of your results example cases, you compare one of the 'reference' datasets with a satellite product, observe differences between the two, then say well they are maybe different because the reference dataset has issues (e.g. related to spatial scales and how they were aggregated) ...so why produce

this reference dataset and use it in the first place? What's the value of a bad reference dataset that we don't really trust?

As the reviewer is surely aware of, we have limited available reference data in the polar regions to support validation of altimetry observations over sea ice (especially for earlier periods, and for specific regions e.g. the Laptev Sea). Therefore, the alternative to many of these cases will be to have no reference data. Furthermore, this depends a lot on the requirements of the users. If one is for example looking to verify thin ice, then the ship data might be appropriate, as the bias towards choosing a route with thinner ice is expected to be smaller if the ice thickness in the region is generally thinner. Understanding and evaluating the limitations of different parts of the reference data is a central element in this study.

Similarly, you treat airborne data as a 'reference' dataset, but I think that is very dangerous. NASA's Operation IceBridge is great for coverage and the multi-sensor nature of the mission, but it still has a lot of issues that are frustratingly yet to be resolved, e.g. the big uncertainties in snow depth from different algorithms applied to the snow radar (King et al., 2015, Kwok et al., 2017) and significant biases between the quick-look and final snow depths (Petty et al., 2023, Fig. S3) which needs to be acknowledged. I was quite surprised this wasn't mentioned at all really.

Thank you for pointing this out. We intentionally use the terms *reference measurements* or *reference observations* instead of *validation data* to acknowledge that reference observations are not necessarily the absolute truth and each comes with its limitations, including the airborne data. We acknowledge that the caveats and limitations of the OIB data are not sufficiently addressed. Additionally, reviewer #2 has notified us that helicopter EM measurements tend to preferentially sample thicker ice, avoiding thin ice and open water for safety reasons. We will adjust the text to highlight these limitations and include the suggested references.

I also think for this study to work, you should try to actually characterize the uncertainties and/or errors in a consistent way. Your effort to summarize how the uncertainties are described in the product is a decent one and I appreciated the effort you put into this. But take IceBridge for example, you neglect all the algorithm differences I point to above, so how useful really are those individual product uncertainties?

We appreciate your acknowledgement of the effort put into summarizing the uncertainties. Uncertainty quantification in individual validation/reference products is an important and pertinent topic being also one of the subjects of lasting discussions on how to proceed with defining this 'reference' data collection and what the reviewer himself wrote earlier. We therefore consider questioning the provided uncertainties to lie beyond the scope of the presented study. Uncertainty characterisation for every single reference measurement used in this study is a significant undertaking, which requires a study in itself. It is also the focus of several FRM projects (e.g., SIN'XS, St3TART/St3TART-FO) and a complete traceable uncertainty characterisation (through either error propagation or Monte Carlo simulations) from initial measurement to the provided

product used in the RRDP requires a thorough evaluation of the different assumptions and methodological steps taken in the initial processing conducted, as well as consideration for correlation, covariance and distributions. For now, we will have to rely on the uncertainties provided in the products, while noting the limitations mentioned by the reviewer (such as representation errors). However, as already mentioned, several projects are currently looking more into this topic (e.g., FDR4ALT, SIN'XS, St3TART-FO, FRM4ALT). However, we will acknowledge the uncertainty introduced by using the different OIB algorithms as also stated in the previous comment.

You state that the reference data should be 'used with care' a few times, but to me this is the job of this study! Decide which data to remove as it is just not a trust-worthy reference dataset for satellite validation for whatever reason. Seems like a cop-out to just say use it with care.

We see your point, however, different reference data is useful for different purposes and removing some data, when the availability is so sparse, also does not seem like the right approach. As suggested above we propose creating a second version of the dataset where we aim to classify the observations with an additional flag to suggest the most useful ones. However, ultimately, "what data is the most useful" is still up for debate when it comes to which data is "best used" to validate satellite observations (e.g., we still need to smooth up to 25 km or more to get reasonable comparisons to radar altimetry, even with airborne data).

Finally, the datasets listed as future work (IceBird, MOSAIC, Nansen Legacy) would have been great to see in this study! Again I think this paper was neither exhaustive of all available data nor thorough in the methodology, so I encourage the authors to decide on a better strategy based on my comments above.

New data become available continuously. The ESA CCI SIT CDR, for which this reference dataset was intended for quality control, is currently only available until 2020. We already agreed to extend the datasets in the reference observation dataset to cover the period of 1993 to the present, including snow depths measurements from the Icebird campaign since 2017. We will consider including reference measurements similar to those already included in the reference dataset e.g., helicopter EM measurements from MOSAiC, Nansen Legacy, and N-ICE upon availability. However, we will not include new methods, such as drone measurements, in the current version, as we would like to keep relatively long consistent time series as these are used for quality control of CDR. Please, let us know if there are specific reference observations you had in mind.

**2 Answer to review by Alek Petty - Specific comments**

I thought it was strange how much the intro talked about radar issues. Why not make it more about the science of why we want to measure basin-scale sea ice thickness? Then if your focus is radar, make that clear from the start,

laser creeping in sometimes was confusing. Probably also easier to reference the papers that discuss the various issues in more detail, keep your focus on the reference datasets.

We agree that the introduction has a strong focus on satellite altimetry methods, which are important for understanding how to handle reference measurements – particularly since the reference dataset was produced as part of the ESA CCI project with the specific aim of being compared with the ESA CCI SIT CDR for consistency. We will update the introduction to place greater emphasis on reference observations. However, we would like to retain the most relevant satellite altimeter considerations in the paper, as they support the discussion section and contribute to understanding the comparability of sea ice variables from reference observations and satellite altimetry. Please also see our responses to the more detailed comments below.

L39 – I think that's still very much TBD and depends on the approach/freeboard used etc!

We agree. Indeed, snow density and snow depth are not the only main contributors depending on instruments, e.g., surface roughness has recently been stated as a significant contributor for the radar-derived altimetry aspects by Landy et al. 2020. We will rephrase this statement for clarity.

L41 – this is mixing up actual errors and theoretical uncertainties propagation which I think is confusing. We agree. We will correct these statements.

L45 – this seems like a bit of a stretch for an introduction! Do we really know that with confidence? Is that true for all types of freeboard and ice regime? None of the statements of this line is stated with certain confidence (e.g., it states that snow load **may** be most important over thin ice, whereas sea ice density *may* be the largest for thicker ice. Furthermore, we also state that it depends on the snow and ice conditions). However, we will revise this section and adapt accordingly.

L47 – well this is really 'a lack of uncertainty quantification data' rather than uncertainties directly I think.

We are not sure what you are referring to here. Can you, please, elaborate?

L80 onwards – ok so your aim is to reconcile radar thickness measurements. I think it would thus help to start with what you interested in then provide the uncertainty discussion to back that up, as before it was confusing how little you talked about laser.

The CDR is SIT, so shouldn't thickness be the main validation target?

The reference dataset was created by CCI to aid and assess the radar altimeter-based SIT CDR. The estimated sea ice thicknesses provided in the CCI SIT CDR are derived from the satellite radar altimetry freeboard measurements. Therefore both freeboard and thickness reference data are necessary. From a metrological point of view, the most accurate comparisons are made by introducing the least uncertainties, e.g., by comparing freeboard reference measurements (whether radar or laser, depending on the sensor used) with freeboards from the satellite CDR. Thus, we aim to use the measurands in their most native form when comparing the reference observations to the satellite altimetry observations, albeit considering thicknesses or freeboard. This is also highlighted in the first paragraph of Section 6.

L135 – "this data package and the methodologies applied herein have the potential of becoming the reference for future comparisons of current and future SIT products." This is a big claim and I don't think you have demonstrated this potential considering all the caveats and issues, and the basic methodology (aggregation) discussed here and even in your results.

We acknowledge that the wording is too strong in this sentence. We propose to change the formulation to:
"This data package and the methodologies applied herein provide initial efforts in collecting, unifying and comparing SIT reference measurements from different reference data sources. The data package and this paper provide a starting point for future work in assessing the uncertainty and reliability of reference measurements."

L407: How is accuracy qualitative? A little confused by that statement. I think it's basically the same as error, no? So it requires a known truth? Whereas uncertainty can be more theoretical.

It is true, that both accuracy and error require a known truth. However, accuracy is qualitative—it describes how close a measurement is to the true value in broad terms (e.g., excellent, good, poor). In contrast, error is a quantitative measure that specifies the deviation between the true value and the measured value (e.g., 10 cm). That said, in practice, the two terms are often used interchangeably despite their distinct statistical meanings. The uncertainty is a parameter characterizing the spread of the quantity values attributed to a measurand. Uncertainty can in plain language be seen as doubt, whereas error can be seen as a mistake as stated by Harris et al. 2017.

L505 ok so maybe stick with the higher number of 10 cm then?

The number of 10 cm refers to an upper limit of the bias, as stated in Lee et al. 2015, and does not necessarily reflect the uncertainty associated with representation error, which, as we have previously stated, is yet to be resolved. Since we acknowledge that we have yet to quantify this uncertainty term, we will consider alternative values such as the 10 cm value proposed here. However, we believe a more appropriate approach would be to further emphasize the issue of representation error in the text and to implement a quality flag, as previously proposed, to indicate data affected by known representation error issues.

L598: "Collocation is performed by finding all satellite data points obtained within $\pm$ 15 days from the date of the reference data, and within the 25 km (50 km for SH) grid cell of the reference coordinates. The average (arithmetic mean) of these satellite points are subsequently allocated to the reference data." Ok so what uncertainties do we think this introduces? I think you need to provide some educated guesses at the very least.

We agree, that we somehow need to address this pertinent issue. This also relates to representativeness and error propagation. As we currently are working on defining the framework of error propagation in other projects (see general comments), we will not be able to fully implement this in the current study. However, we will suggest to investigate the representativeness; for example, if

there is only one available flight, the average over a month would not be as representative as e.g., for the moorings, where both the reference and satellite measurement are expected to have the same temporal averaging. However, here we still assume that the sea ice covering the mooring due to the sea ice drift over a month is equivalent to the sea ice covered by a satellite within a grid cell, even though the mooring is permanently fixed to the same location. We suggest making a sensitivity study where we change the time averaging to include e.g., $\pm 7$ days or even $\pm 3.5$ days of the satellite CDR around the time-stamp of the reference measurements to see the impact of these for the different types of reference measurements. This would allow us to provide the study with a statistical basis for temporal representativeness. However, by using this approach we do not take into account that the satellite data is not equally distributed over a month either, which would also introduce another uncertainty. The above-mentioned approach could go along with a more extensive discussion of the representation and the uncertainties introduced by using our method in sections 7.5 and 7.6

L660 – why bother comparing if you then say it's not right to compare them? Would you have stated the same if the stats were better? Much better to state from the off which data are appropriate to compare against and why, then show how to use those..!

We see your point. However, polar reference observations are very sparse and we do not have data that match the spatial and temporal scales of satellite altimetry. Therefore, we prefer to retain the reference observations currently included in the dataset. If we do not perform some form of statistical comparison, even with data that have limited use for satellite CDR quality control, we cannot determine which reference measurements are appropriate to use. We here propose to clarify how and where different reference data sources are most appropriately used. This will be supported by our proposed dataset update, which introduces flags to categorize reference observations based on our interpretation of the statistical results according to their suitability for satellite CDR quality control. For example, ship-based observations and IMBs would rank low, and could be filtered out by users through these flags. To support our discussions we believe it is important to retain all currently included data, even those with representativeness issues, since it is undeniable used in scientific work. We will make a critical assessment of the reference data, cite relevant literature using these data for validation, and align the manuscript with this new approach i.e., implementing flags to categorize data according to their usability.

IMB discussion – ok so there's two things – you're underestimating the actual uncertainties AND also not really dealing with the representation error.

We agree and acknowledge that we need to further consider how to address these two issues. Since IMBs measure localized thermodynamic growth, we believe the best solution, as previously proposed, is to implement flagging to highlight the representativeness issue of these measurements, which makes them unsuitable for comparisons with data on a 25/50 km scale. We are, however, very keen to know if you have any suggestions for us on how to implement this. Specifically, in terms of the uncertainty contribution due to representation error

(which to our knowledge is not well described in papers on satellite altimetry sea ice thickness or in comparisons to reference measurements) and in terms of quantifying the actual uncertainty. Do you have a study in mind?

"Additionally, no specific uncertainty for SD versus SIT is provided, resulting in the acoustic rangefinder sounders' accuracy used as the uncertainty for both SD and SIT." Why? I think you should be attempting to figure out what that should be, even if you have to make some assumptions.

We agree to look more into this and will make a dedicated section or paragraph in the manuscript, which discusses this in more detail. The uncertainty of the sensor retrieval, which includes the initial snow depth, sensor tilt and undetected snow-ice formation, is only one thing. We lack studies investigating the typical uncertainty of measurements of initial snow depth, which according to Nicolaus and Katlein 2017 is the primary source of uncertainty in snow depth measurements itself. The much bigger issue is the representativeness of the target variable. A realistic uncertainty assumption likely needs to be in the order of at least 20 to 30 cm. We cannot think of a single altimetry thickness study that also takes into account the representation issue (whether it be inequal thickness distribution from localised buoy measurements compared to satellites, different coverage of grid cell by both satellite and buoys etc.), and we will make sure to discuss this in the manuscript. Are you aware of any such studies or do you have an alternative idea of how we can address this uncertainty?

**References**

Lee, J. E. et al. (2015). "Uncertainty Analysis for Evaluating the Accuracy of Snow Depth Measurements". In: *Hydrology and Earth System Sciences Discussions* 12.4, pp. 4157–4190. DOI: 10.5194/hessd-12-4157-2015.

Harris, I. et al. (2017). "Updated high-resolution grids of monthly climatic observations – the CRU TS3.10 Dataset". In: *Earth System Science Data* 9.1, pp. 511–527. DOI: 10.5194/essd-9-511-2017. URL: https://essd.copernicus.org/articles/9/511/2017/.

Nicolaus, M. and C. Katlein (2017). "Observations of the Snow Depth on Arctic Sea Ice". In: *Journal of Geophysical Research: Oceans* 122.9, pp. 7167–7183. DOI: 10.1002/2017JC012838.

Landy, Jack C. et al. (2020). "Sea Ice Roughness Overlooked as a Key Source of Uncertainty in CryoSat-2 Ice Freeboard Retrievals". In: *Journal of Geophysical Research: Oceans* 125.5. e2019JC015820 2019JC015820, e2019JC015820. DOI: https://doi.org/10.1029/2019JC015820. eprint: https://agupubs.onlinelibrary.wiley.com/doi/pdf/10.1029/2019JC015820. URL: https://agupubs.onlinelibrary.wiley.com/doi/abs/10.1029/2019JC015820.

Jutila, Arttu et al. (2022). "High-Resolution Snow Depth on Arctic Sea Ice From Low-Altitude Airborne Microwave Radar Data". In: *IEEE Transactions on Geoscience and Remote Sensing* 60, pp. 1–16. DOI: 10.1109/TGRS.2021.3063756.

Da Silva, Elodie et al. (2023). "Towards Operational Fiducial Reference Measurement (FRM) Data for the Calibration and Validation of the Sentinel-3 Surface Topography Mission over Inland Waters, Sea Ice, and Land Ice". In: *Remote Sensing* 15.19. ISSN: 2072-4292. DOI: 10.3390/rs15194826. URL: https://www.mdpi.com/2072-4292/15/19/4826.

Ida Olsen and Henriette Skourup (Feb. 2025). *ESA-CCI-RRDP-code*. Version v1.0. DOI: https://doi.org/10.5281/zenodo.14808969. URL: https://github.com/Ida2750/ESA-CCI-RRDP-code.

---

## Author Comment (AC2)

**Response to referee**

**ilo and hsk**

**January 2025**

**1 Answer to review by anonymous - General review**

The manuscript aims to combine a range of "reference" dataset containing sea ice thickness, sea ice freeboard, snow depth and sea ice draft measurements. It's commendable the amount of work that has gone into compiling this dataset. However, the dataset and the description of it requires some additional work. Specifics has already been pointed out by Reviewer 1 and I have opted to not repeat those also here.

We appreciate the reviewer's time and feedback, which have helped identify areas for improvement. We acknowledge the concerns raised and will make the necessary revisions to improve clarity, consistency, and accuracy. Below are our detailed responses to each comment. Please, see also our replies to reviewer #1

**2 Comments**

A concern for the quality of the reference dataset is the introduction of unknown information into the reference dataset, e.g., by adding a timestamp to the AEM-AWI when a time is missing. The adding of a time stamp (effectively assigning the data additional information) is then listed as a very minor pre-processing. Similarly, is the submarine dataset manipulated by adding dates to the dataset, the pre-processing level is then listed as 1-3. Minor pre-processing in my opinion would be removing clearly erogenous data points in a QC of the data, not adding information. A definition of the authors interpretation of the pre-processing flags is therefore needed in the manuscript. For section 8 should it be clearly stated that at times NaN have been replaced with arbitrary time stamps.

We will clarify the use of pre-processing flags to ensure sharper and more precise definitions and we propose to add this information in a table for better overview. Regarding the AEM-AWI data, the additional information provided is a timestamp within the given day, meaning the maximum error introduced is 24 hours. Given that the data is later averaged on a monthly scale, the impact of this addition is minimal. However, we acknowledge the need to explicitly state this in the manuscript to avoid confusion.

The IMB-CRREL data is listed as the most important data in this study (L337-338) and has also been listed as a dataset where major pre-processing is needed, and that the data should be used with care (L342). Assigning the dataset as the most important and at the same time the most unreliable raises questions to the usefulness of this entire reference dataset.

Agreed. However, the statement on L337-338 appear misleading. The intended meaning was that mass balance data (snow depth and sea ice thickness) are more relevant to our study than temperature measurements (available in the buoy data). We will revise the wording accordingly.

In section 3 the levels 0-3 are listed as pre-processing flags, and in section 4 levels 0-3 are listed as uncertainty flags. Using the same numbers/listings may cause confusion, a recommendation is therefore to use different numbers for the different types of levels.

Thank you for raising this point, we acknowledge that using the same numerical values (0-3) for both pre-processing and uncertainty flags may be confusing. However, as they are provided as two separate variables, they should be straightforward to implement and separate from one another, when using the data. This is a common use of flags (e.g., ICESat-2 implements several quality flags with values ranging from 0-2 or more). We will clarify in the text, with use of acronyms (QF = Quality Flag, UF = Uncertainty Flag), which flag the specific value is specified for.

How was the length of the different sensor time series selected? E.g., why is the Fram Strait wide mooring data after 2018 not included in the study (L224), why are IMB buoys only until 2015 included (L242), and why are only ASSIST data until 2021 included (L278)? On L238, why is the updated version of the data not used in this study? The updated dataset is from 2022 which is now 3 years ago.

We thank the reviewer for this comment. In terms of the Fram Strait wide mooring data, the data source from which we have gotten the data only contains data until 2018-08-31 (Sumata, Divine, and Steur 2021). All IMB data from after 2015 lacks measurements of snow depth and sea ice thickness. The latest deployed buoy was the 2016A buoy in the Beaufort Sea but this buoy, unfortunately, contains no measurements of snow depth and sea ice thickness (Perovich, Richter-Menge, and Polashenski 2022). We propose to extend time-series, such as the ASSIST data, and include data of similar types, such as HEM from MOSAiC, Nansen and N-ICE, but not include entirely new data types (see also responses to reviewer #1).

Deformed ice appears to be defined as ice >3m thick (L441). Meaning that large parts of MYI will fall into the deformed ice category and many FYI areas, incl. rubble fields, will be classified as level with this definition. Should perhaps a different thickness range have been used here? Rough ice is used in other places (e.g. L462), what is the overall sea ice classifications used in this work? Should rough be the same as deformed? For clarity please include a sea ice type definition early on in the manuscript, I understand that there may be a large number of ice types and that there will be significant overlap between different ice types but it's good to be able to see the definition in one place.

This is a great recommendation. We acknowledge that the use of rough, deformed and thick sea ice interchangeably is confusing and this will be revised accordingly.

Section 4.2.4. This data is based on visual observation and is very dependent on the experience of the person making the observations, yet this dataset has been given an uncertainty flag of 1. Whereas data that are independent on human errors such as the airborne data has been given an uncertainty flag of 3. Measured and quantified errors have therefore been given a higher degree of uncertainty than those who is not easy (impossible?) to assess errors. E.g. how common is it for the estimates for the ship data to be made by an experienced observer? In addition is there a statement on L367-369 where the ship-observations show the opposing trend to what is expected, is the reason behind this the data quality? On L669 the SIT for ship observations is described by the authors as dubious. Should the ship-based observations perhaps therefore be ranked a level 3 like the airborne measurements? Section 7.5. It is great to see uncertainty in the data being discussed, though a clear and consistent definition is needed.

These are valid concerns. In terms of the ship data, we believe that the opposing trend, e.g. that the sea ice thickness measured by ships tends to be lower than the sea ice thickness measured by satellites, is mainly caused by a bias in ship routes toward thinner ice. Regarding your point about the uncertainty flagging, we have currently used uncertainty flags as a way to quantify, the level of variability and information that we have in our uncertainty, but as you point out we lack to quantify information such as representation error and human interpretation in our flagging. We will look into making updated uncertainty flag categories, to take into account these. However, differences in data sources make it challenging to establish a fully consistent flagging method across all datasets without introducing a degree of subjectivity. We encourage suggestions to specific datasets, which you believe should have different uncertainty or pre-processing flags.

Section 7.1. Flight measurements may at times avoid certain ice types, such as thin/young ice, and open water areas which will affect the sea ice distribution in these datasets. The deployment of IMBs on pre-dominantly stable (thicker) sea ice is brought up it would be useful to also discuss the un-representativeness of the ice types in the air-borne data. This may help explain the discrepancies discussed on L638-640. The sea ice in the brackish Baltic Sea has a very low salinity, in areas equivalent to fresh ice. What is the error uncertainty associated with this ice type in the reference dataset?

Thank you for raising this. We were not aware of the fact that flight measurements may at times avoid certain ice types, and have not encountered this when considering flight lines below satellite orbits where the main aim is to capture the same ice conditions as observed by the satellites. We have since been made aware that single-engine helicopters may avoid flying at low level over water. However, missing open water is not expected to have a large impact since the CCI sea ice thickness is the "mean thickness of the ice-covered fraction of the grid cell area". Nevertheless, this is an interesting point, that we will look

further into and we will make sure to raise this point in the manuscript when we discuss representativeness.

A "reference" dataset such as the one mentioned could be used by other research groups outside the altimeter community. The down sampling to 25 km NH (50 km for SH) grid cell of the reference coordinates, therefore, makes the compiled dataset less useful. Others within the altimeter community may also want to perform this down sampling in different mathematical ways, potentially rendering the dataset less used than if the original resolution of the reference data is kept. The effect of this down sampling should also be discussed with the uncertainty assessment in mind, i.e. what uncertainty is introduced by the down sampling. A quantification of the uncertainties and errors introduced to the data should be discussed and quantified in section 7.6.

As we have made the processing pipeline (code on GitHub available Ida Olsen and Henriette Skourup 2025 ), where it is relatively easy for the user to change time-space averaging to their needs. We will at this stage not be able to quantify the errors introduced by the time-space averaging as this is ongoing research, see also general comments to reviewer #1. However, we propose to include additional information in the paper about how the code can be modified to be used for other purposes, to make it more straightforward for readers to obtain the reference data in the desired temporal and spatial resolutions. Publishing the data in its original resolution amounts to redistributing the source data and this is not in the scope of this work. However, links to all source data are provided in table 2, making it straightforward for potential users to obtain the used source data in is original resolutions.

**3  Minor comments**

L4-5. What is meant with format matching? The file type, the data type?

We acknowledge that this is not very well formulated. We here refer to time-space averaging. We will update the text accordingly.

L47-48. The issues with combining in-situ/airborne/drone based etc validation data to any remote sensing sensor is challenging due to the differences in temporal and spatial resolution. The snow depth is not unique in this regard, and it is unclear why this parameter is listed as uniquely challenging.

Here, we justify why we include snow depth reference measurements in the RRDP, even though it might not be the prime measurand or even derived from the dedicated satellite altimetry mission. Specifically, snow depth plays a pivotal role when converting into sea ice thickness from satellite (or airborne) altimetry, and therefore a comparison with reference observations of other aspects of the sea ice (its thickness or draft) will rely on the snow depth product used in this conversion. However, we have been encouraged by reviewer #1 to update the introduction with more focus on the reference observations. During this process, we will update these sentences for clarity.

L138. Why is section 8 listed before 3,4... etc?

Thank you for bringing our attention to this. This will, of course, be adjusted to keep the sections in chronological order.

Table 1. ASSIST data originates not only from ice breakers, but also ice capable ships etc. Later in the text the terms ship (e.g. Figure 2) or support vessels are used, it would be good to be consistent throughout the manuscript. Ship would suffice.

Thank you for raising this point, we will use ship throughout the manuscript for consistency.

Figure 2. How was dominate data source assigned? >8%?

If you refer to the numbers in bold between campaign/provider and geophysical variable in Figure 2, then these numbers were highlighted as they show the primary contributor (whichever is highest) to each of the four main geophysical variables (SD, SIT, FRB and SID). We will clarify this in the caption.

Introduce the acronyms the first time they are used, e.g. now AEM is used many times not explaining that is stands for Airborne Electro Magnetic soundings (?) EM is explained on L184.

Thank you for pointing this out. We will ensure that acronyms are described when first mentioned.

L337-338. What makes this the most important data for this study?

As previously mentioned, this is a mistake in the wording and it will be updated.

L451-452. What is the uncertainty for the NPI data?

The uncertainty for NPI data is described in lines 453 to 458 and is stated in Table 4.

L467. Consider using months instead of summer and winter to allow for easier interpretation of the time period for the SH and NH.

Thank you for the suggestion. We will implement this approach.

L476. Is depth = draft depth?

Yes it is. We will here correct "depth" to "draft depths" when necessary.

**References**

Sumata, Hiroshi, Dmitry Divine, and Laura de Steur (2021). *Monthly mean sea ice draft from the Fram Strait Arctic Outflow Observatory since 1990.* DOI: 10.21334/npolar.2021.5b717274. URL: https://doi.org/10.21334/npolar.2021.5b717274.

Perovich, D., J Richter-Menge, and C Polashenski (2022). *Observing and understanding climate change: Monitoring the mass balance, motion, and thickness of Arctic sea ice.* URL: http://imb-crrel-dartmouth.org.

Ida Olsen and Henriette Skourup (Feb. 2025). *ESA-CCI-RRDP-code.* Version v1.0. DOI: https://doi.org/10.5281/zenodo.14808969. URL: https://github.com/Ida2750/ESA-CCI-RRDP-code.

---

## Author Comment (AC3)

**Response to community comment**

ilo and hsk

January 2025

We thank Dr. Robbie Mallett for taking the time to comment on our paper and contribute to the discussion. This is greatly appreciated.

**1 Answer to community comment by Robbie Mallett**

Given what Alek has written about significant inter-product variability in snowradar data, I wanted to briefly raise a point about line 635; it's suggested that snowradar-derived radar freeboards & snow depths can be used to "directly evaluate" the penetration depth of CryoSat-2's SIRAL instrument.

We tried to do exactly this for some recent work, and found that the derived penetration depth depended quite strongly on the snowradar algorithm, such that we could not meaningfully achieve what the authors are suggesting in L635.

Indeed; we fully agree, and this was an oversight. Actually, during the exploration of data presented in recent work under review (Fredensborg Hansen et al. 2024), we also tried evaluating penetration depth using CReSIS snow radar (although not presented in the paper). However, we came to the same conclusion as you - that it depends on the snow radar algorithm since the snow depth from that radar is used as the "true snow depth", and since there it yet to be a convincing case of the "best" product, this is not trivial.

Our investigation is documented in Section 2 of the supplementary material of Nab et al. (2024). The authors are right that if there were some roughly constant CS2 "penetration factor", then it could be estimated by regressing the difference in the CS2 & OIB radar freeboards against the coincident OIB-derived snow depths. Higher snow depths would lead to bigger mismatches in the radar freeboards, as the impact of limited penetration would grow. The rate at which the mismatch scales with snow depth would reveal the penetration factor: if the mismatch remained the same as the snow got deeper, the CS2 penetration factor would be 100%. If the mismatch grew in a 1:1 ratio with the snow depth, then the inferred CS2 penetration would be zero (i.e. operating as a laser altimeter).

When we did this, we found the regression slope is 0.21 (penetration = 80%) for the QL product, but 0.6 (penetration = 40%) for the wavelet and peakiness retrackers deployed with pysnowradar. So we couldn't estimate the penetration depth in this way, without assuming one algorithm is so good as to

be "the truth". It's possible that there is a "best" algorithm, but I'm yet to see a convincing case made.

Agreed. Also here, the inter-comparison of (Kwok et al. 2017) highlights the differences in snow radar algorithms.

There is an alternative way of doing this where you assume penetration happens by absolute (not fractional) depth. I.e. Let's imagine the CS2 return originates X cm below the snow surface, vs X % of the snow depth below. This approach also leads to an unacceptable level of variability in the derived penetration depth based on snowradar algorithm.

For what it's worth, our inability to figure out the CS2 penetration depth with snowradar data led to our use of ULS moorings in the main part of the paper. The ULS data allowed us to calculate some penetration depths a bit more reliably. If the authors are looking for a way in which the reference measurements compiled here allow us to learn about CS2 penetration depths/factors, this is potentially a good example.

Nab, C., Mallett, R., Nelson, C., Stroeve, J., & Tsamados, M. (2024). Optimising interannual sea ice thickness variability retrieved from CryoSat-2. Geophysical Research Letters, 51(21), e2024GL111071.

Thank you for raising this point and for pointing us towards your very interesting paper. We completely agree with your observations and will revise our statement on line 635 accordingly, and incorporate references to your work (Nab et al.). We agree that your use of ULS data, rather than OIB data, serves as an interesting case study demonstrating the importance of discussing reference data limitations. In line with our response to reviewer #1, we will expand our discussion on the limitations of OIB data, particularly highlighting differences between snow retrieval algorithms.

**References**

Kwok, R. et al. (2017). "Intercomparison of snow depth retrievals over Arctic sea ice from radar data acquired by Operation IceBridge". In: *The Cryosphere* 11.6, pp. 2571–2593. DOI: 10.5194/tc-11-2571-2017. URL: https://tc.copernicus.org/articles/11/2571/2017/.

Fredensborg Hansen, R. M. et al. (2024). "Exploring microwave penetration into snow on Antarctic summer sea ice along CryoSat-2 and ICESat-2 (CRYO2ICE) orbit from multi-frequency air- and spaceborne altimetry". In: *EGUsphere* 2024, pp. 1–53. DOI: 10.5194/egusphere-2024-2854. URL: https://egusphere.copernicus.org/preprints/2024/egusphere-2024-2854/.

---

## Author Comment (AC4)

**Response to editor**

**ilo and hsk**

**March 2025**

Thank you for your responses to the reviewers and community comments. After reviewing your responses, we believe the paper is on the right track but will require major revisions to fully address the comments. Below, we outline areas where additional work is needed:

Thank you very much for your editorial comment, this is highly appreciated. We agree with your assessment and appreciate the concrete suggestions for solving some of the issues addressed by the reviewers. We will implement the suggested changes and submit a revised manuscript. Please, find below the concrete comments related to the points highlighted.

**1 Response to Reviewer 1 (Alek Petty):**

Reviewer 1 queries the use of the term 'reference'. While you propose a revised title, we recommend removing 'reference' from the title. Similarly, it would be helpful to take care with the terms 'reference measurements' or 'reference observations' to ensure that users of the dataset clearly understand the limitations. We acknowledge that the term "reference measurements/observations" should be clearly defined. However, we would prefer to keep the term and define its meaning at the very beginning of the paper. The term "reference measurements" serves as an alternative to "validation measurements" (which would be incorrect due to data limitations) and "in-situ measurements" (which typically do not include remote measurements such as airborne campaigns). The Copernicus service itself refers to all "reference" observations as in-situ, stating: "Within Copernicus, in-situ data refers to local, on-site or in-position observation data collected from ground, sea, or air-borne sensors as well as geospatial reference data, imagery gathered by drones (Unmanned Aerial Vehicles or UAVs) and information collected by crowds of volunteer contributors." (`https://insitu.copernicus.eu/about/what-is-in-situ-data`). However, we believe this definition does not align with the general understanding of "in situ," which typically refers to on-ground measurements. For readability, we prefer to use a common term that refers to the collection of non-satellite measurements. We therefore propose to continue using the term reference observations/measurements (depending on whether they are actual observations or measurements) in the paper, including in the title. However, we will state clearly

in the beginning of the paper what we define as reference measurements e.g. a collection of non-satellite measurements (in-situ and airborne), which can be used as a comparison to satellite remote sensing measurements. If you have an alternative term, which we can use, we would appreciate suggestions.

We appreciate the paragraph from your response highlighting this as a "first approach" and emphasising the broader community usage of the methodology. We recommend this is clearly articulated in both the introduction and discussion sections. This will help contextualise the work and address Reviewer 1's broader concerns. Consider reorganising the introduction to highlight the broader goals of the study before delving into technical challenges.

Thank you for the suggestion. We will make sure to highlight that this is a "first approach" and further emphasize the broader community usage of the methodology first in the introduction and during the discussion section.

Reviewer 1 highlights the importance of addressing uncertainties. While redoing uncertainty analyses for each dataset is out of scope, it is essential to ensure that users of the dataset are aware of the validation processes relating to the underlying datasets. In each case, we suggest:

- explicitly stating the community standards or validation processes that the input datasets have undergone including a critical discussion of the limitations of these datasets and their potential impact on the results

- In particular, the discussion on airborne data (Operation IceBridge) requires more explicit acknowledgement of algorithmic differences and their impact on uncertainties, as suggested by the reviewer.

We will make sure to revise relevant sections ensuring that they include an overview of validation procedures used in the production of the input datasets. For OIB in particular we will make sure to highlight algorithmic differences, along with uncertainties related to these, including adding relevant references to existing literature.

Reviewer 1 expressed concerns about the inclusion of data that may not be reliable for satellite validation due to issues like spatial and temporal mismatches or other uncertainties. While you propose flagging data for representativeness issues, this must be implemented clearly in the dataset and supported with clear definitions and examples of how to use the flags to ensure that users can make informed decisions.

We will ensure that the flagging process is supported with clear definitions of the flagging methodology. All flags will be consolidated into a single table with adequate descriptions. Additionally, we will include examples illustrating the application of these flags, such as demonstrating the effects of removing data with representativeness issues and discussing the impact of this approach.

Reviewer 1's comment on L598 about collocation and representation errors is significant. While you acknowledge this issue, it is not sufficient to defer it to future work. The reviewer suggests that this method introduces uncertainties related to temporal and spatial representation errors, which should at least be

estimated or discussed in the study. To address the reviewer's concerns, we recommend

- Including a preliminary analysis or educated estimates of the introduced uncertainties

- Discussing how these uncertainties could affect the results.

We will look further into this matter to decide how we can best address this point. As an initial step we propose to do a sensitivity study, where we examine the impact on the matchup statistics if we use other temporal matchup windows, such as one week. We will ensure that we discuss how uncertainty linked to representation errors could affect the results.

You posed a number of questions to Reviewer 1 in your response. As he has not responded on the Copernicus interactive discussion interface, we recommend that you contact him directly to clarify these points.

We will contact him directly regarding remaining questions.

**2 Response to Reviewer 2 (Anonymous Referee):**

Reviewer 2 raises valid concerns about the pre-processing and uncertainty flags. While you have proposed clarifications, the manuscript should include a clear table or section explicitly defining the flagging system and examples of how users should interpret these flags.

We will include a table with clear definitions of the flags, and provide examples of how the users should inpret these flags.

The reviewer also questions the inclusion of certain datasets with significant pre-processing (e.g., AEM-AWI timestamps, IMB-CRREL data). Your response defends these choices but does not fully address the broader implications for data reliability. Please elaborate further in the manuscript to justify its importance explicitly and discuss its limitations.

We will clarify this in the manuscript.

The issue of inconsistent time series lengths (e.g., Fram Strait data ending in 2018, IMB data until 2015) should be addressed more explicitly. If newer data is unavailable, this should be clearly stated in the manuscript. Where possible, it would be useful to indicate when updates might be made.

We will ensure that the extent of the available datasets is clearly stated, along with how/when/if they are expected to be updated and extended.

Your response to the specific reviewer comment on L47-48 does not directly address the reviewer's point. You could explicitly state that while snow depth is not uniquely challenging compared to other parameters, it is highlighted due to its critical role in sea ice thickness estimation.

Thank you for the suggestion. We will update the text accordingly.

**3 Community Comment (Robbie Mallett):**

The comment about the variability in penetration depth estimates from snow radar algorithms is critical. While you acknowledge this limitation in your response, the manuscript should explicitly discuss the limitations of OIB data and their influence on uncertainties. Ensure that the discussion of penetration depth variability is improved and supported by relevant citations.

We will include a paragraph to elaborate on the uncertainties linked to the use of OIB data, both regarding the algorithm uncertainties and the uncertainties linked to penetration depth variability. We will support this with relevant citations.

**4 Comments common to both reviewers:**

Both reviewers emphasized the need for a more consistent and transparent approach to uncertainty quantification. Reviewer 1 and 2 both highlight the need to address representation errors more thoroughly. Your proposal to flag data for representativeness issues is a good start, but it should be supported with quantitative examples or case studies in the manuscript.

We will ensure to both flag data with known representativeness issues and to include examples of how to use them. We propose conducting case studies to assess the statistical impact of including and excluding data with known representation errors when comparing with satellite altimetry data.

Both reviewers note the absence of key datasets (e.g., IceBird, MOSAiC, Nansen Legacy). While you propose including these in future updates, the manuscript should include a clear roadmap for how the dataset will be expanded and improved over time. We also encourage the inclusion of available, long-term consistent timeseries, as you mention in the last paragraph of Section 1 in your response to Reviewer 1.

We have extended the data record to include the latest available data of all data sources and we have included IceBird data. We will include some of the MOSAiC data and the Nansen Legacy data, which is similar in methods to already existing data e.g., drifting buoys and helicopter/airborne EM. We will provide a concrete plan for how and when we expect to update the database.

We encourage you to proceed with the revisions and submit a revised manuscript for further consideration.

Thank you, this is highly appreciated.